# Active carpets drive non-equilibrium diffusion and enhanced molecular fluxes

Francisca Guzmán-Lastra[1,2,5✉], Hartmut Löwen[3] & Arnold J. T. M. Mathijssen [4,5✉]

Biological activity is often highly concentrated on surfaces, across the scales from molecular motors and ciliary arrays to sessile and motile organisms. These 'active carpets' locally inject energy into their surrounding fluid. Whereas Fick's laws of diffusion are established near equilibrium, it is unclear how to solve non-equilibrium transport driven by such boundary-actuated fluctuations. Here, we derive the enhanced diffusivity of molecules or passive particles as a function of distance from an active carpet. Following Schnitzer's telegraph model, we then cast these results into generalised Fick's laws. Two archetypal problems are solved using these laws: First, considering sedimentation towards an active carpet, we find a self-cleaning effect where surface-driven fluctuations can repel particles. Second, considering diffusion from a source to an active sink, say nutrient capture by suspension feeders, we find a large molecular flux compared to thermal diffusion. Hence, our results could elucidate certain non-equilibrium properties of active coating materials and life at interfaces.

[1] Escuela de Data Science, Facultad de Estudios Interdisciplinarios, Universidad Mayor, Santiago, Chile. [2] Departamento de Física, FCFM Universidad de Chile, Santiago, Chile. [3] Institut für Theoretische Physik II: Weiche Materie, Heinrich-Heine-Universität Düsseldorf, Düsseldorf, Germany. [4] Department of Physics & Astronomy, University of Pennsylvania, Philadelphia, PA, USA. [5] These authors contributed equally: Francisca Guzmán-Lastra, Arnold J. T. M. Mathijssen. ✉email: franciscaglastra@gmail.com; amaths@upenn.edu

magine a particle floating in outer space, surrounded by uniformly distributed stars that all exert a gravitational force on it. On average, the total force is zero, by symmetry, but its variance is infinite, as described by the Holtsmark distribution[1]. Microscopically, a similar situation occurs when molecular motors or swimming cells generate long-ranged flows and fluctuations in 'living fluids'[2–12]. These active suspensions operate far from thermal equilibrium where surprising effects emerge that fundamentally impact biological processes, including enhanced diffusion of enzymes[13–15], viscous and chaotic mixing by motility[16–23], viral infections[24], bioconvection[25–27], oxygen redistribution[28], nutrient uptake[29–32] and communication via hydrodynamic trigger waves[33].

Besides active suspensions, biological activity is commonly concentrated on surfaces. Hence, 'active carpets' can form that drive systems out of equilibrium by injecting energy from boundaries. These carpets exist across the length scales: Inside cells, cytoskeletal motors generate cytoplasmic streaming while membranes host catalysing enzymes and transport proteins, all producing non-equilibrium fluctuations and active stresses[34–40]. Outside cells, ciliary arrays create globally directed flows across entire organs but locally also facilitate mixing[41–47], which together may enhance pathogen clearance[48]. Cells themselves accumulate on surfaces too, driving flows to replenish nutrients, including biofilms or microbial colonies[49–53] and sessile suspension feeders[54–58]. Multicellular organisms also drive feeding currents from walls, including sponges, reef corals[59,60], carpets of upside-down *Cassiopea* jellyfish[61] and other macrobenthos[62] at larger Reynolds numbers. Beyond biological hotspots, synthetic active carpets have been developed, including artificial cilia[63,64], self-propelled droplets and colloids[9,10,65,66], engineered bacterial carpets[67,68], molecular motility assays[69], light-controlled microfluidic flow networks[70], hydrogel actuators, liquid crystal elastomers and other responsive materials[71–73].

In this article, the properties of non-equilibrium diffusion of molecules or passive particles are examined near such active carpets that generate long-ranged flows. We distinguish between various types of carpets that inject energy into the surrounding medium in different ways: We consider actuators that exert a net force on the liquid (like pumping cilia or sessile suspension feeders), a net torque (like mixing rotors), or a net stress (like swimming bacteria). First, we show how the resulting active fluctuations decay with distance from the surface. Next, we derive the space-dependent diffusivities for the different carpet types, verified by hydrodynamic simulations, and the corresponding generalised Fick's laws are written out. These laws are then used to solve the active analogue of two classic problems: We first examine sedimentation of particles towards an active carpet. Instead of following the Boltzmann distribution, the particles can be repelled by the fluctuations close to the surface, so the surface features an active self-cleaning effect. We then consider the diffusion of molecules (or particles) from a source to an active sink, such as a surface covered with sessile suspension feeders. The nutrient flux (or particle capture rate) is found to scale quadratically with the active forcing, so it can be significantly larger than the flux due to thermal diffusion. Overall, these problems can be described by relatively simple equations, even if the resulting dynamics feature unexpected solutions.

## Results

### Active carpet definition

We consider an active carpet made of actuators that generate flows at low Reynolds numbers near a planar no-slip surface (Fig. 1a–d). The surface is located at $z = 0$. The actuators have position $\boldsymbol{r}_a = (x_a, y_a, h)$ and orientation unit vector $\boldsymbol{p}_a$. Each actuator $a$ drives an individual flow, $\boldsymbol{u}(\boldsymbol{r}, \boldsymbol{r}_a, \boldsymbol{p}_a)$,

evaluated at position $\boldsymbol{r}$. In general, this flow can be written in terms of the Blake tensor $\mathcal{B}(\boldsymbol{r}, \boldsymbol{r}_a)$[74] and a multipole expansion thereof (see 'Methods: Individual flow fields').

For example, molecular motors walking or cells crawling along a substrate can entrain the surrounding fluid (Fig. 1a)[34–37,39,40]. This can be described by a 'parallel Stokeslet', a point force $f_\parallel$ oriented parallel to the surface, giving rise to the flow $\boldsymbol{u}_\parallel = f_\parallel \mathcal{B} \cdot \boldsymbol{p}_a$. At a larger scale, sessile suspension feeders like *Vorticella* cells generate nutrient currents (Fig. 1b)[54,55], which can be described by a point force $f_\perp$ oriented perpendicular towards the surface, $\boldsymbol{u}_\perp = f_\perp \mathcal{B} \cdot \boldsymbol{e}_z$. Similarly, the flow generated by an *E. coli* bacterium swimming parallel to a wall (Fig. 1c) is described well[50] by a 'Stokes dipole' flow, given by $\boldsymbol{u}_D = \kappa (\boldsymbol{p}_a \cdot \nabla_a)(\mathcal{B} \cdot \boldsymbol{p}_a)$, with dipole moment $\kappa$. Finally, a torque-generating actuator can be described by a Stokes rotlet (Fig. 1d), given by $\boldsymbol{u}_R = \varrho((\boldsymbol{e}_x \cdot \nabla_a)(\mathcal{B} \cdot \boldsymbol{e}_y) - (\boldsymbol{e}_y \cdot \nabla_a)(\mathcal{B} \cdot \boldsymbol{e}_x))/2$, with strength $\varrho$. This could represent rotors on a surface, such as bacteria with tethered flagella, or a carpet of nodal cilia[45] that move around in circles in the $xy$ plane. A more complex example could be (non-nodal) airway cilia, beating almost perfectly in a plane but with some off-plane fluctuations[48,75]. For such situations, to establish a more realistic description, one can combine terms from this multipole expansion, using Stokeslets, rotlets, dipoles and higher-order terms as needed.

To be explicit, we have written out the full expressions of these first multipoles in 'Methods: Individual flow fields', and their corresponding flow fields are illustrated in Fig. 1e–h. Also note, throughout this paper we consider non-dimensional quantities in our simulations and figures, but all results can be predicted by analytical expressions, into which dimensional values can be inserted. Hence, we will discuss our results for typical numbers in biology.

Once the individual flows $\boldsymbol{u}$ are established, the total flow $\boldsymbol{v} = \sum_a \boldsymbol{u}$ due to all $N_a$ actuators is probed by a passive tracer particle as a function of its distance from the surface. For a given carpet architecture, which is defined by the probability density $F(\boldsymbol{r}_a, \boldsymbol{p}_a)$ of finding an actuator at position $\boldsymbol{r}_a$ and orientation $\boldsymbol{p}_a$, the average total flow evaluated at position $\boldsymbol{r}$ is $\langle \boldsymbol{v}(\boldsymbol{r}) \rangle = \int \boldsymbol{u} F \mathrm{d}\boldsymbol{r}_a \mathrm{d}\boldsymbol{p}_a$, where $\langle \ldots \rangle$ denotes averaging over a statistical ensemble of independent active carpet configurations. If there are any spatial or orientational gradients in the distribution $F$, for example due to bacterial clustering, or topological defect patterns, then long-ranged flows can emerge[52]. However, in this paper we will consider cases where the actuators are uniformly distributed, so $F$ is equal to a constant. Then the mean drift cancels out, so $\langle \boldsymbol{v} \rangle = \boldsymbol{0}$ in the absence of gradients in the carpet architecture.

### Active fluctuations

Even if the mean flow generated by the actuators is equal to zero, its variance at any one time is not. Hence, the flows can lead to 'active fluctuations' that push and pull on particles near the carpet. We first determine the strength of these fluctuations numerically. A carpet is simulated by placing $N_a$ actuators that are randomly distributed with a uniform surface density $n$ within the $xy$ plane. We then evaluate the total flow $\boldsymbol{v}$ evaluated at particle position $\boldsymbol{r}_0 = (0, 0, z_0)$. By repeating this for a large ensemble of $N_e$ independent carpet configurations, we evaluate the distribution of the total flow, PDF($\boldsymbol{v}$), analogous to the Holtsmark distribution[1]. Subsequently, the moments of this total flow distribution (the mean, variance, skewness and kurtosis) are found as a function of distance from the surface, for different carpet types (see 'Methods: Characterising fluctuations: simulation details').

We first focus on carpets made of Stokeslets oriented parallel to the surface, with uniformly distributed orientations, so $p_z = 0$ and $F = n/2\pi$. Figure 1i shows the resulting histogram of the total

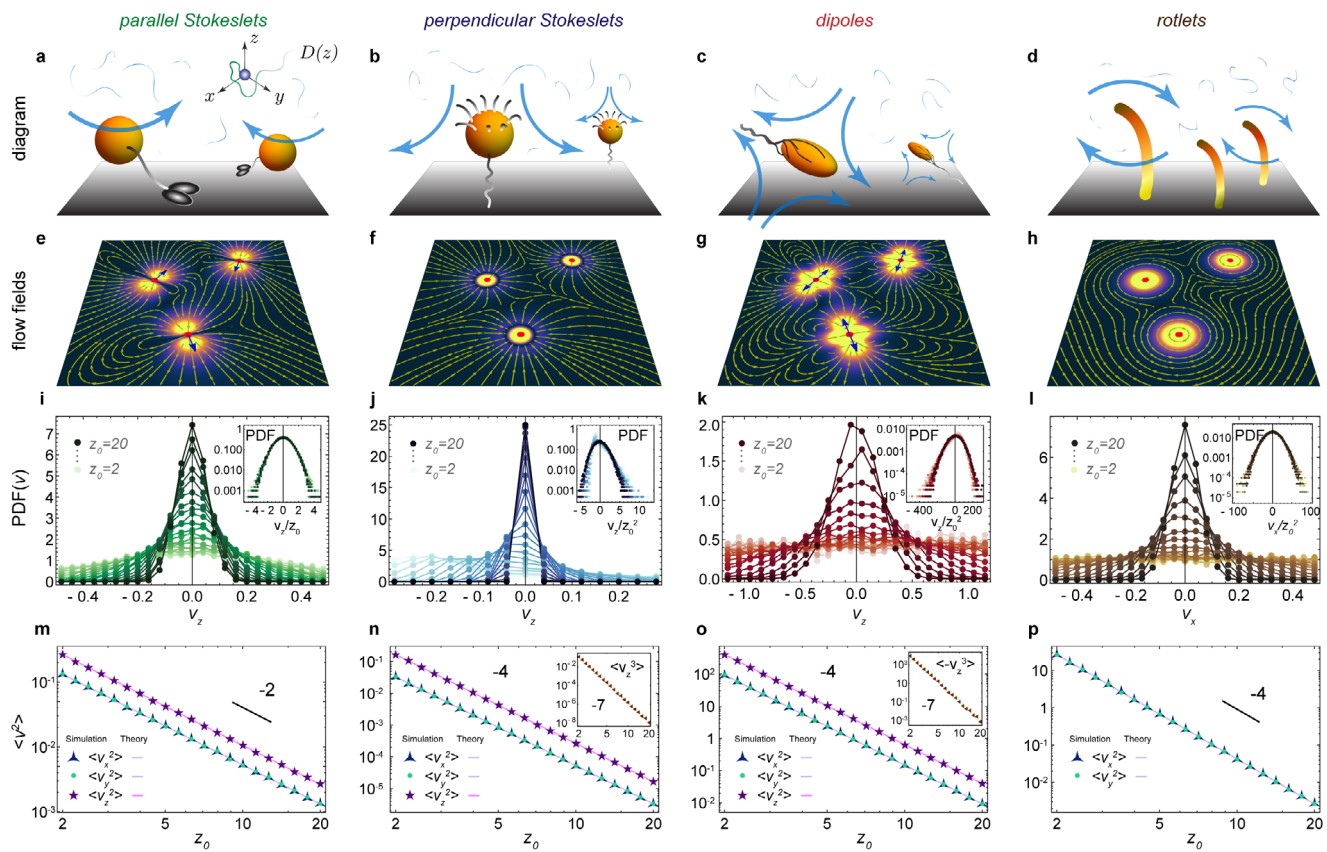

**Fig. 1 Flows driven by different types of active carpets.** Diagrams of a few examples: **a** Molecular motors walking along a surface, described by parallel Stokeslets. **b** Sessile suspension feeders that draw in feeding currents, described by perpendicular Stokeslets. **c** *E. coli* bacteria swimming along a surface, described by Stokes dipoles. **d** Actuators that rotate about the $z$-axis, such as nodal cilia, described by Stokes rotlets. **e–h** Flow fields generated by these different actuator types, shown in the $xy$ plane at $z_0 = 2h$. **i–l** Probability distribution functions, PDF($v_z$), of the total flow velocity due to $N_a = 10^5$ actuators, at different heights $z_0$. Insets show rescaled histograms that highlight skewness and kurtosis, especially for small $z_0$ values. **m–p** The variance of these distributions, $\langle v_i^2 \rangle$, corresponding to the strength of active fluctuations in different directions $i$, as a function of distance $z_0$ from the active carpet. Insets show the skewness for **n** and **o**, but this is zero for **m** and **p**. Symbols show simulation results and the lines are the theoretical predictions of Eq. (1) and Table 1.

vertical flow, $v_z$, for different values of $z_0$. This distribution is an even function, by symmetry of the parallel Stokeslet, so it is immediately clear that the mean flow vanishes. However, the width is finite and decays with distance from the surface, so the active fluctuations are stronger closer to the carpet. This variance can be calculated analytically,

$$\mathcal{V}_{ij} = \langle v_i v_j \rangle = \int u_i u_j F \mathrm{d}\boldsymbol{r}_a \mathrm{d}\boldsymbol{p}_a, \qquad (1)$$

for the directions $i$, $j \in x$, $y$, $z$, as detailed in 'Methods: Characterising fluctuations: theory details'. For the vertical component, for instance, this integral yields the variance

$$\langle v_z^2 \rangle = 6\pi n h^2 f_\parallel^2 / z_0^2. \qquad (2)$$

Physically, this expression represents how deeply the active boundary can influence the passive bulk fluid. This theoretical result is listed in Table 1 (row 4, column 3), and it is compared with simulations in Fig. 1m (magenta line). The decay with distance is $1/z_0^2$ for all diagonal components, but the off-diagonal components are zero. Interestingly, the variance in the vertical direction (purple stars) is twice as strong as in the horizontal directions (green points, blue triangles), so $\langle v_x^2 \rangle = \langle v_y^2 \rangle = \frac{1}{2} \langle v_z^2 \rangle$ for parallel Stokeslets.

Moreover, when repeating this algebra for Stokeslets oriented perpendicular to the surface, we find that $\langle v_z^2 \rangle$ is five times stronger than $\langle v_x^2 \rangle$ (Table 1, fourth column) and the decay is now

$1/z_0^4$. This scaling also holds for the dipole flows (fifth column) and the rotlets (sixth column). Note that rotlets only generate flows in the $xy$ plane, so their resulting variance only has horizontal components. This fact could be exploited to tune the relative magnitude of $\langle v_x^2 \rangle$ and $\langle v_z^2 \rangle$. For example, one could use an active carpet made of both Stokeslets and rotlets and vary their relative prefactors $f_\perp$ and $\varrho$, or vary their relative densities $n_\perp$ and $n_\varrho$. Therefore, as we discuss next, the diffusion anisotropy may become a tunable.

The distributions of the active fluctuations, PDF($\boldsymbol{v}$), are not purely Gaussian. They feature skewness and kurtosis (Table 1; bottom rows), but they still obey the central limit theorem because the variance is finite for $z_0 > 0$. Otherwise, Lévy flights must be considered[12,16,76,77]. Additionally, these higher-order moments also decay with $z_0$, so far away from the active carpet the profiles are more Gaussian (Fig. 1i–l; insets).

**Space-dependent diffusivity**. To understand how these active fluctuations may lead to particle diffusion, it is important to realise how exactly they vary over time. Most systems in nature are dynamic, such as bacteria (or synthetic self-propelled particles) that swim over a surface, with some source of stochasticity. As the bacteria move and reorient, the flows they produce change dynamically. Therefore, a particle advected by these flows will trace out a path that is eventually diffusive. As opposed to Brownian motion, though, this diffusion process is not the sum of

**Table 1 Properties of fluctuations driven by an active carpet.**

| | $\gamma$ | Parallel Stokeslets | Perpendicular Stokeslets | Parallel dipoles | Rotlets |
|---|---|---|---|---|---|
| $u(r; \mathbf{r}_a, \mathbf{p}_a)$ | | $\mathbf{u}_\parallel$, Eq. (14) | $\mathbf{u}_\perp$, Eq. (15) | $\mathbf{u}_D$, Eq. (16) | $\mathbf{u}_R$, Eq. (17) |
| Mean | 1 | (0, 0, 0) | (0, 0, 0) | (0, 0, 0) | (0, 0, 0) |
| Variance | 2 | $(1, 1, 2)\frac{3\pi n h^2 f_\parallel^2}{z_0^2}$ | $(1, 1, 5)\frac{3\pi n h^4 f_\perp^2}{z_0^4}$ | $(7, 7, 30)\frac{3\pi n h^2 \kappa^2}{4z_0^4}$ | $(1, 1, 0)\frac{3\pi n h^2 \rho^2}{2z_0^4}$ |
| Skewness | 3 | (0, 0, 0) | $\left(0, 0, \frac{483840}{4199}\right)\frac{\pi n h^6 f_\perp^3}{z_0^7}$ | $\left(0, 0, -\frac{552960}{4199}\right)\frac{\pi n h^3 \kappa^3}{z_0^7}$ | (0, 0, 0) |
| Kurtosis | 4 | $(3, 3, 20)\frac{54\pi n h^4 f_\parallel^4}{35z_0^6}$ | $(267, 267, 12,880)\frac{81\pi n h^8 f_\perp^4}{1001z_0^{10}}$ | $(2797, 2797, 107,680)\frac{27\pi n h^4 \kappa^4}{2288z_0^{10}}$ | $(1, 1, 0)\frac{27\pi n h^4 \rho^4}{14z_0^{10}}$ |

The columns correspond to different types of hydrodynamic actuators: we consider point forces (Stokeslets) oriented parallel to the surface, perpendicular Stokeslets, parallel dipoles, and rotlets. The rows correspond to different moments of the total flow velocity: we consider the mean, the variance, the skewness, and the kurtosis of these flows, presented as $(\langle v_x^\gamma \rangle, \langle v_y^\gamma \rangle, \langle v_z^\gamma \rangle)$ using the exponent $\gamma = 1$, 2, 3, 4 respectively, as a function of distance from the carpet, $z_0$. Note that the off-diagonal components are equal to zero in all cases.

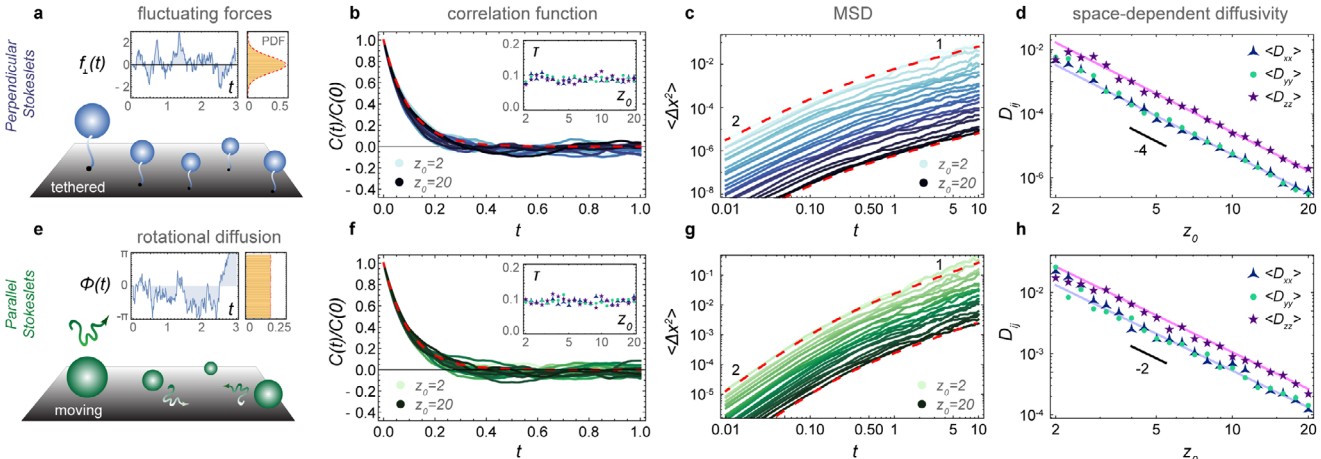

**Fig. 2 Diffusion driven by an active carpet.** For **a**–**d** the surface is covered with sessile perpendicular Stokeslets with independent random forces. Also see Supplementary Movie 1. For **e**–**h** the surface is covered by parallel Stokeslets moving along a surface with a constant velocity $V$ and rotational diffusion $D_r$. For both cases we simulate the motion of tracer particles subject to these active fluctuations, ensemble-averaged over $N_e = 100$ independent tracer trajectories, as a function of their initial distance from the surface, $z_0$. **a** Diagram and typical time course of a random force $f(t)$ described by an Ornstein–Uhlenbeck process. **b** Velocity correlation function (VCF) of the tracer velocity over time. Blue lines show different values of $z_0$, each ensemble-averaged, and the red dashed line is the prediction of Eq. (48). The inset shows that the resulting correlation time $\tau$ is independent of $z_0$. **c** Mean-squared displacement (MSD) for different values of $z_0$, each ensemble-averaged, transitioning from ballistic to diffusive motion. The red dashed lines show the theoretical approximation of Eq. (3) for $z_0 = 2, 20$. **d** Anisotropic diffusivity in the directions parallel (blue, green) and perpendicular (purple) to the surface. The solid lines show the prediction of Eq. (4). **e** Diagram and typical time course of the orientation angle $\phi_a(t)$ described by rotational diffusion. **f**–**h** are equivalent to **b**–**d** for moving actuators.

random kicks that are instantaneous. Instead, the active fluctuations are correlated in time. They have a memory based on the history of the active carpet configuration.

Therefore, we first focus on a simple case with only one memory time scale, $\tau$. Consider a carpet of sessile suspension feeders like *Vorticella convallaria*, tethered organisms that produce time-varying nutrient currents (Fig. 2a). These are modelled as $N_a$ perpendicular Stokeslets that are again uniformly distributed in space, and fixed because they are non-motile. Each organism generates a flow with its own time-varying strength $f_\perp = f(t)$, which is independent of the other organisms. These $N_a$ forces fluctuate according to an Ornstein–Uhlenbeck process, $\frac{df}{dt} = -f/\tau + \sigma\eta(t)$, with relaxation time $\tau$ and strength $\sigma$. These behavioural quantities are set by the cells' intrinsic properties, such as the internal biochemistry and biophysics. Supplementary Movie 1 depicts these actuator dynamics as well as the evolution of the total flow $\mathbf{v}(t)$ they produce. We then simulate the motion of tracer particles that do not diffuse because of Brownian thermal fluctuations, but because of the fluctuating flows generated by the active carpet (see 'Methods: Simulating the diffusivity of particles near carpets made of fluctuating forces').

The resulting velocity correlation function (VCF) and mean-squared displacement (MSD) are shown in Fig. 2b and c, respectively. The tracer dynamics are ballistic at short times, $t \ll \tau$, called the Holtsmark regime. However, at long times they are diffusive with a linear relation $\langle \Delta x^2 \rangle = 2D_{xx}t$, and similarly in the other directions. Hence, we determine the components of the diffusion tensor as a function of $z_0$ (Fig. 2d). Like the flow variances, the vertical component $D_{zz}$ is five times stronger than the horizontal components, leading to anisotropic diffusion, and they all decay as $1/z_0^4$ in all three directions.

This system can be solved analytically when considering the limit of small displacements, $\langle \Delta r^2 \rangle \ll r_0^2$, when the noise amplitude is small. This ensures that we determine the local diffusivity, $D_{ij}(z)$ with small variations in $z$. Using information from the variance of the active fluctuations and their temporal correlations, the motion can then be integrated (see 'Methods: Derivation of the mean-squared displacement and space-dependent diffusivity'). This gives the MSD,

$$\langle \Delta r_i \Delta r_j \rangle = 2\mathcal{V}_{ij}\tau\left(t + \tau(e^{-t/\tau} - 1)\right), \quad (3)$$

which captures both the short-term ballistic motion and the diffusivity after long times (Fig. 2c, dashed red lines). Thus, for the vertical Stokeslets for example, we find the space-dependent diffusion,

$$D_{zz}(z_0) = 15\pi n h^4 \langle f^2 \rangle \tau / z_0^4 \\ = 5D_{xx} = 5D_{yy}, \tag{4}$$

which is compared with the simulations in Fig. 2d. Because our theoretical approximation is formulated for small amplitudes of active fluctuations, the expression only holds far from the surface, when $z \gg \sqrt[6]{30\pi n h^4 \langle f^2 \rangle \tau^2}$ (Eq. (54)). Beyond this distance we find a good agreement between the simulations and the theory.

Overall, we find that the diffusion driven by an active carpet can be much stronger than Brownian thermal diffusion. Considering a carpet of *Vorticella* with cell radius $a \sim 25\,\mu m$, $h \sim 150\,\mu m$, $n \sim 1/(100\,\mu m)^2$, $\tau \sim 1\,s$ and $8\pi\mu f_\perp \sim 500\,pN$[55], the active diffusion can be dominant up to distances of $z_{th} = \sqrt[4]{15\pi n h^4 \langle f^2 \rangle \tau / D_{th}} \sim 1\,mm$ for small nutrient molecules of $D_{th} \sim 500\,\mu m^2/s$. That is much larger than the cell size. Moreover, for micron-sized prey of $D_{th} \sim 0.5\,\mu m^2/s$, we find $z_{th} \sim 7\,mm$, orders of magnitude larger than the organism itself. Hence, we expect these results to be highly relevant across the scales, for non-equilibrium transport from molecular to organismic sizes.

**Diffusion due to motile actuators**. These concepts can be extended to a more complex system with motile actuators (Fig. 2e–h). We consider Stokeslets that move with a constant velocity $V$ along the surface, subject to rotational diffusion $D_r$, which exert on the liquid a constant force $f_\parallel$ that is aligned with the direction of motion (see 'Methods: Simulating the diffusivity of particles near carpets made of moving actuators'). Importantly, this system features multiple time scales: The reorientation time, $\tau_r \sim 1/D_r$, and the time taken to move underneath a particle, $\tau_u \sim z_0/V$. The corresponding tracer dynamics are therefore more complicated to solve in general, but for slow actuators the decorrelation of the carpet memory is primarily controlled by the rotational diffusion, since $\tau_r \ll \tau_u$ for small $V$. Indeed, as shown in Fig. 2f, in that case the VCF simulated for different positions $z_0$ (green lines) closely follows the actuators' rotational correlation function (red dashed). Then the same theory as before can be applied to predict the MSD (Fig. 2g) and the diffusivity (Fig. 2h). In particular, we can approximate the space-dependent diffusion for parallel Stokeslets as

$$D_{zz}(z_0) = 6\pi n h^2 f_\parallel^2 / D_r z_0^2 \\ = 2D_{xx} = 2D_{yy}, \tag{5}$$

and similarly for other actuator types following Table 1. In the remainder of this paper we stay in the small $V$ limit, but future work should also address the case of faster actuators.

Again, the active diffusion is significant compared to thermal diffusion. Inserting some typical values into Eq. (5), say $h \sim 1\,\mu m$, $n \sim 1\,\mu m^{-2}$, $D_r \sim 1\,s^{-1}$ and $8\pi\mu f \sim 1\,pN$, we have $z_{th} = \sqrt{6\pi n h^2 f_\parallel^2 / D_r D_{th}} \sim 7.7\,\mu m$ for molecules of $D_{th} \sim 500\,\mu m^2/s$. Similarly, for micron-sized particles with $D_{th} \sim 0.5\,\mu m^2/s$ we have $z_{th} \sim 240\,\mu m$, so again much larger than $h$. Since $nh^2$ remains constant when scaling down in size, the enhanced transport can be significant even for subcellular systems.

**Generalised Fick's laws**. Once the local diffusivity is determined, we can establish the generalised Fick's laws (or Fokker-Planck equations) that govern the global stochastic transport. For clarity, we first revise the case of constant diffusion $\tilde{D}$. Fick's first law relates gradients in particle concentration $\varphi(\mathbf{r}, t)$ and an external drift flow $\mathbf{v}_d$ to the total flux, $\mathbf{J} = \mathbf{v}_d \varphi - \tilde{D}\nabla\varphi$. Fick's second law describes how the particle concentration evolves in time, $\partial_t \varphi = -\nabla \cdot (\mathbf{v}_d \varphi) + \tilde{D}\nabla^2\varphi$. This follows directly from the first law and the continuity equation, $\partial_t \varphi = -\nabla \cdot \mathbf{J}$.

If the diffusivity is not constant, the spatial dependence may either arise from gradients in the mean speed or an inhomogeneous mean free path[78]. Indeed, as we saw earlier (Eqs. (4) and (5)), the diffusion tensor $D_{ij}(z) = \mathcal{V}_{ij}(z)\tau(z)$ can be written in terms of the variance of the active fluctuations $\mathcal{V}$ and the memory time $\tau$, which in general can both depend on position. As described in 'Methods: Generalised Fick's laws', the diffusion driven by an active carpet can be analysed by constructing a 'telegraph model', following Schnitzer[78]. The particle flux is then described by the first generalised Fick's law,

$$\mathbf{J}(\mathbf{r}) = \mathbf{v}_d\varphi - (\tau\mathcal{V} \cdot \nabla)\varphi - \frac{\varphi\tau}{2}\nabla \cdot \mathcal{V}, \tag{6}$$

and the second generalised Fick's law still follows from the continuity equation. Interestingly, these laws can be used to describe the active analogue of several classic problems, as we discuss in the next sections.

**Sedimentation towards an active carpet**. As a first application, we consider particles sedimenting towards an active carpet (Fig. 3a, 'Methods: Sedimentation towards an active carpet: simulation details'). The fluctuations are driven by slowly moving parallel Stokeslets, as before. Typical particle trajectories $z(t)$ are shown for different values of $v_g$, the sedimentation velocity (Fig. 3b). As expected, the particles with the smallest $v_g$ reach the highest $z$ positions (green track) but, surprisingly, they hover above the active carpet at some finite height. This phenomenon is somewhat reminiscent of the Leidenfrost effect, where droplets hover above a hot plate. That is, far from the active carpet the particles fall under gravity, but nearby they are repelled by strong active fluctuations (also see Supplementary Movie 2). Because the system is driven out of equilibrium, the sedimentation profile $\varphi(z)$ does not follow the Boltzmann distribution with an exponential decay. Instead, it features a maximum value (Fig. 3c) with much less particles close to the surface than expected.

To quantify this 'self-cleaning' effect, we solve the generalised Fick's laws for this system analytically. For active fluctuations that scale algebraically with distance from the carpet, $\mathcal{V}(z) = \tilde{\mathcal{V}}/z^\alpha$ as in Table 1, and similarly for the memory time, $\tau(z) = \tilde{\tau}/z^\beta$, the vertical component of the generalised flux (Eq. (6)) becomes

$$J_z(z) = -v_g\varphi - \frac{\tilde{D}}{z^{\alpha+\beta}}\frac{\partial\varphi}{\partial z} + \frac{\alpha\tilde{D}}{2z^{\alpha+\beta+1}}\varphi, \tag{7}$$

where the constant part of the vertical diffusivity is $\tilde{D} = \tilde{\mathcal{V}}_{zz}\tilde{\tau}$. We require that $J_z = 0$ at steady state, which yields the non-Boltzmannian sedimentation profile

$$\frac{\varphi(z)}{\varphi_0} = z^{\alpha/2} \exp\left(-\frac{v_g z^{\alpha+\beta+1}}{\tilde{D}(\alpha+\beta+1)}\right), \tag{8}$$

where $\varphi_0$ is a normalisation factor. To clarify, $v_g$ has standard units of m/s and $\tilde{D}$ has units of $m^{2+\alpha+\beta}/s$. In the limit of a constant diffusivity ($\alpha = \beta = 0$) we recover the Boltzmann distribution, $\varphi(z) = \varphi_0 e^{-v_g z/\tilde{D}}$. A major different with the sedimentation profile near an active carpet is the $z^{\alpha/2}$ factor in Eq. (8), where $\alpha = 2$ for parallel Stokeslets. Therefore, the sedimentation profile features a maximum (Fig. 3c), which is

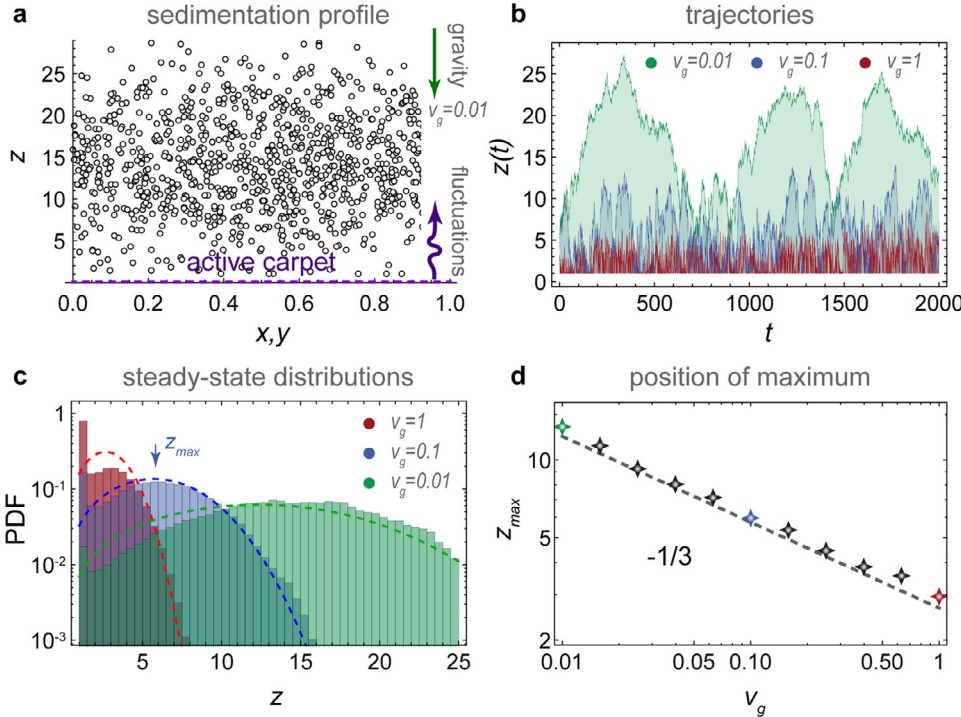

**Fig. 3 Sedimentation towards an active carpet.** Particles sink with velocity $v_g$ due to gravity, but are pushed up by the active fluctuations generated by the carpet (dashed purple line), which in this case is composed of moving parallel Stokeslets. **a** Diagram showing the steady-state sedimentation profile. **b** Typical trajectories $z(t)$ for a range of sedimentation velocities $v_g$. **c** Steady-state distributions of particle density for different $v_g$. Histograms show simulations and dashed lines are the prediction of Eq. (8). **d** Position of the maximum particle density as a function of $v_g$. Markers show simulations and the dashed line is Eq. (9).

located at

$$z_{\max} = \left( \alpha \tilde{D} / 2 v_g \right)^{1/(\alpha+\beta+1)}. \tag{9}$$

These results agree with our simulations (Fig. 3c, d), for different values of the sedimentation velocity.

The self-cleaning effect can be understood by analysing the generalised flux more carefully (see 'Methods: Self-cleaning effect'). The first two terms are identical to the case of constant diffusion. However, a third term emerges that represents a flux toward regions where the speed is low. Since the active fluctuations decay with distance, this flux term is always directed away from the active carpet. Importantly, the self-cleaning effect also persists when thermal fluctuations are included in the analysis (see 'Methods: Sedimentation with active and thermal diffusion').

**Diffusion from a source to an active carpet sink.** Next, we consider particles diffusing from a source to an active carpet sink (Fig. 4a, 'Methods: Diffusion from a source to an active carpet sink: simulation details'), which could represent particle capture by sessile suspension feeders, or substrate molecules being catalysed by a carpet of enzymes (also see Supplementary Movie 3). Every time a simulated particle diffuses across the absorbing boundary at $z_{\text{sink}} = h$, we place it back at $z_{\text{source}} = H$. This way, a diffusive flux emerges towards the carpet, which is equivalent to the particle capture rate. Typical trajectories $z(t)$ show that the particles spend the majority of time close to the top surface (Fig. 4a), which is also apparent in the concentration profiles $\varphi(z)$ that are peaked at the top surface (Fig. 4b). Moreover, the mean first-passage time is much larger for distant sources, so the flux

decays rapidly with gap size (Fig. 4c), which may be important for biology or applications in confined spaces.

This system can again be solved analytically (see 'Methods: Diffusion from a source to an active carpet sink: theory details'). The flux is given by Eq. (7), with $v_g = 0$ and with fixed particle concentrations at the source and sink. Hence, we must solve the continuity equation $\partial_z J_z = 0$ subject to the boundary conditions $\varphi(H) = \varphi_+$ and $\varphi(h) = 0$. This gives the solution

$$\frac{\varphi(z)}{\varphi_+} = \frac{z^{\alpha+\beta+1} - h^{\beta+1}(hz)^{\alpha/2}}{H^{\alpha+\beta+1} - h^{\beta+1}(hH)^{\alpha/2}}, \tag{10}$$

for $\alpha \geq 0$ and $\beta \geq -1$, or a slightly more complex function for other values. The corresponding solution for the flux, equivalent to the particle capture rate, is

$$\frac{J_z}{\varphi_+} = - \frac{(\alpha + 2\beta + 2)\tilde{D}}{2\left( H^{\alpha+\beta+1} - h^{\beta+1}(hH)^{\alpha/2} \right)}. \tag{11}$$

In the limit of constant diffusion and $h \ll H$, we recover the profile $\varphi(z) \approx \varphi_+ z/H$, which increases linearly from sink to source, and the flux $J_z \approx -\varphi_+ \tilde{D}/H$. However, for an active carpet the profile is more concentrated at the source, in agreement with the simulations (Fig. 4b). Indeed, the flux also decays more rapidly than thermal diffusion with the gap size (Fig. 4c). These predictions (dashed lines) agree well with the simulations.

Finally, we also examine the effect of different force strengths $f_\parallel$ on the diffusion to the active sink. Interestingly, we find that the flux scales quadratically with the active fluctuations, so stronger Stokeslets lead to a much larger capture rate (Fig. 4d). Inserting the typical values written below Eq. (5) into Eq. (11), with source concentration $\varphi_+ \sim 1$ particle/$\mu$m and gap size $H \sim 10$ $\mu$m, we find

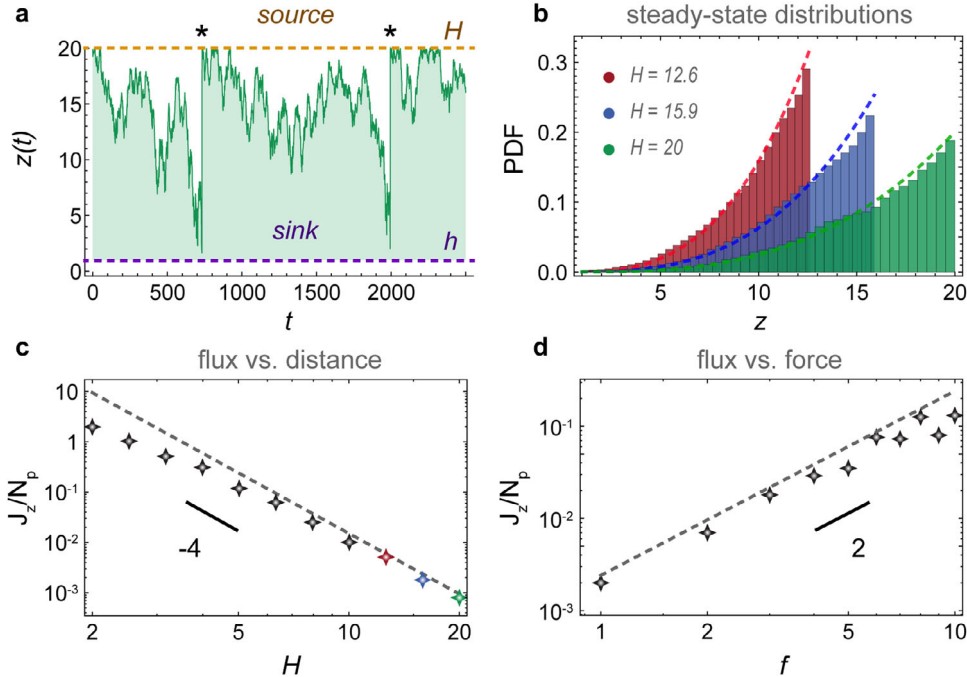

**Fig. 4 Diffusion from a source to an active sink.** Particles are spawned at a source (orange line) and subsequently diffuse to a sink (purple line) due to active fluctuations generated by the carpet, which is composed in this case of moving parallel Stokeslets. **a** Typical trajectory $z(t)$ of a particle. Every time it hits the sink, we place it back at the source (events marked with *). **b** Steady-state distributions of the particle density for different source positions. Histograms show simulations and dashed lines are Eq. (10). **c** Diffusive flux as a function of distance. **d** Diffusive flux as a function of active fluctuation strength. For both **c** and **d** the markers show simulations and dashed lines show the prediction of Eq. (82).

an active flux of $|J_z| \sim 60\,\mathrm{s}^{-1}$. That is more than the thermal flux for micron-sized and molecular particles, $|J_z| \sim 0.05$ and $50\,\mathrm{s}^{-1}$, respectively. Interestingly, the thermal and active fluctuations can co-operate to give an even larger flux of $\sim 128\,\mathrm{s}^{-1}$ together ('Methods: Diffusion from a source to an active carpet sink: comparison with thermal diffusion'). Hence, by enhancing the local diffusivity, the active carpet may increase its nutrient uptake significantly.

**Advective and diffusive transport**. Until now, we have considered active carpets that are spatially uniform, where $F(\boldsymbol{r}_a, \boldsymbol{p}_a)$ is constant so the average flow vanishes, $\langle \boldsymbol{v} \rangle = 0$. However, any natural carpet is likely to feature some heterogeneity in its force distribution that can drive local advection flows (see Methods: Advective and diffusive transport'). This imposes a constraint on the generalised Fick's laws we discussed so far: If the particles are stuck in local advection currents, they cannot diffuse around freely. Interestingly, this is also related to the occurrence of quenched disorder ('Methods: Quenched disorder'). The key question is then how strong these advection flows are compared to the actively driven diffusion.

To investigate this, we consider a carpet composed of a square lattice of perpendicular Stokeslets that all fluctuate about a non-zero mean force $\bar{f}$ with variance $\mathrm{Var}(f)$. The actuators are located at $(i\lambda, j\lambda, h=1)$ where $i, j$ are integer numbers and $\lambda$ is the lattice spacing. The total flow driven by these actuators is then described by an advective contribution and a diffusive contribution, $\boldsymbol{v} = \boldsymbol{v}_{\mathrm{adv}} + \boldsymbol{v}_{\mathrm{diff}}$. As shown in Fig. 5, and described in detail in 'Methods: Advective and diffusive transport', the active diffusion is stronger than the advection beyond a specific distance from the carpet (Eq. (92)). As mentioned earlier, we also require $z$ to be large (Eq. (54)) in order for the time scale of diffusive transport to be slow compared to the time scale of the active fluctuations

themselves. Indeed, this is the case in many biological and engineered settings, but care should be taken that these conditions are satisfied.

## Discussion

In summary, active carpets are ubiquitous in nature, from molecular and cellular to organismic length scales. In this paper we described how these active carpets can drive non-equilibrium fluctuations by locally injecting energy into their surrounding fluid. These fluctuations were quantified with a general theoretical framework in terms of the fundamental solutions of the Stokes equation. Hence, we derived the diffusivity as a function of distance from an active carpet, and we found the corresponding generalised Fick's laws. The predictive power of these laws was demonstrated by solving them for two archetypal problems: sedimentation and diffusion towards an active sink. Of course, one could extend this straightforwardly to other problems like diffusion away from an active source, potentially in combination with sedimentation, or systems with multiple active boundary conditions in one or more dimensions.

This framework can be applied to a much more general class of problems. Until now we have considered non-interacting actuators, where the fluctuating forces and associated time scales are determined by their intrinsic properties. It would be interesting to consider actuators featuring different types of collective behaviours[2–11], say swarming and clustering, so these forcing properties might emerge collectively. This could lead to interesting types of anomalous diffusion[79].

Additionally, one could consider active transport and diffusion in complex geometries (see 'Methods: Extension to more complex geometries'). Immediate extensions could be carpets on the outside of a sphere, such as cilia covering *Volvox carteri*[30] and microbes covering marine snow[80,81], but also carpets on the

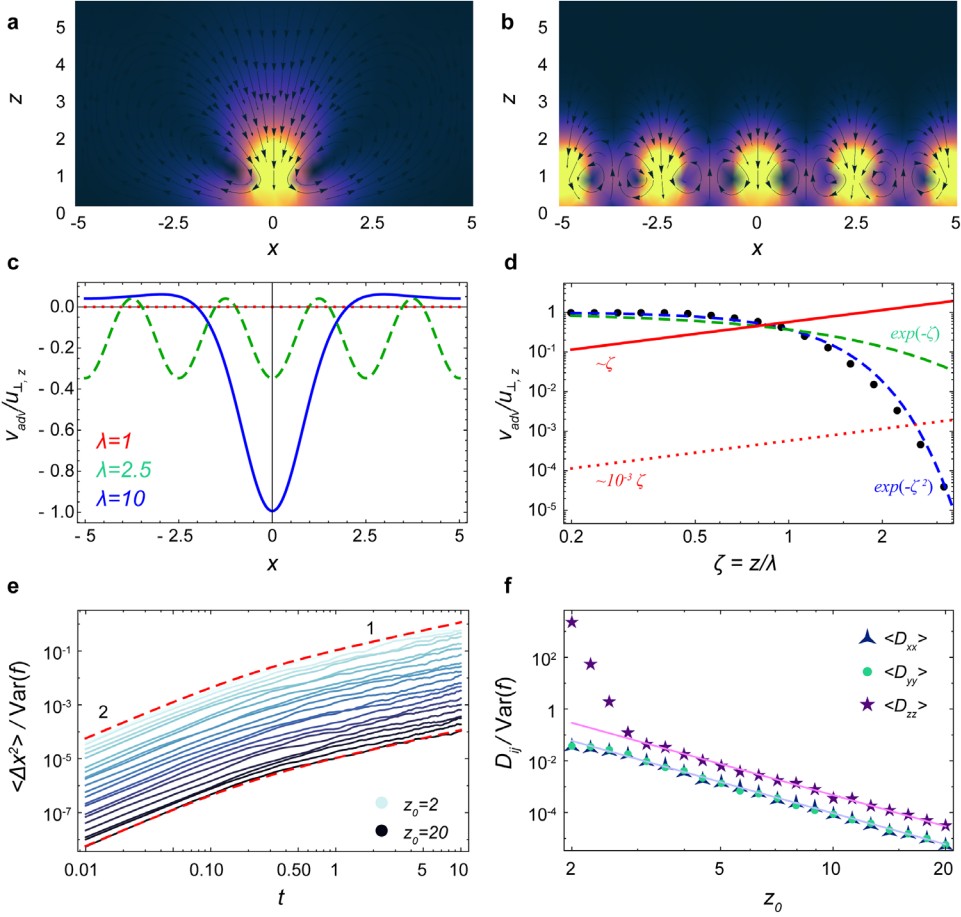

**Fig. 5 Comparison of advective and diffusive transport.** Here the active carpet is a square lattice of perpendicular Stokeslets. **a, b** Flows (Eq. (88)) produced by a carpet of $\lambda = 10$ (sparse) and $\lambda = 2.5$ (dense), respectively, in the plane $y = 0$. Colours shows the flow magnitude and black arrows are streamlines. **c** The total advection flow, $v_{adv,z}(x, 0, 2.5)$, normalised by the flow of a single actuator, $u_{\perp,z}(0, 0, 2.5)$, for different lattice spacings: $\lambda = 10$ (blue), $\lambda = 2.5$ (green), $\lambda = 1$ (red). The flows vanish as $\lambda$ decreases. **d** Comparison of advective and active diffusive transport. Black points show the normalised advection flow, $\Phi(\zeta)$ (see Eq. (89)). The dashed lines are the decaying functions $e^{-\zeta}$ (dashed green) and $e^{-\zeta^2}$ (dashed blue), so the advection vanishes for dense carpets and large distances from the surface. However, the normalised diffusive transport (Eq. (90)) increases with $\zeta$ (red lines). **e** Mean square displacement (MSD) of a particle near a carpet with very weak active fluctuations compared to a strong mean force, $\mathrm{Var}(f) = 10^{-6}$ and $\bar{f} = -1$. The other parameters used are density $n = 1$, $h = 1$, $\tau = 0.1$. The red dashed lines show the prediction of Eq. (3). **f** Corresponding space-dependent diffusivity. The solid lines show the prediction of Eq. (4). Despite the strong advection currents near the carpet, the theory still holds beyond a certain distance from the surface.

inside of a spherical cavity, such as motors on the cytoskeleton[11]. Once these topologies are understood, one could consider more complex curved spaces, like the highly folded active membranes of the endoplasmic reticulum. Furthermore, besides translational diffusion one could derive the rotational diffusion[82,83], for spherical but also for ellipsoidal particles and other shapes[84]. Finally, flows with finite inertia could be modelled to describe carpets with actuators that are large or perform rapid motions[33,85].

Besides hydrodynamic actuators, this framework could equally be applied to different kinds of energy injection from surfaces. For example, one could consider active carpets that generate fluctuating electric or magnetic fields, or catalyse chemical reactions. Conversely, there are also surface-driven thermo-osmotic and acoustic fluctuations, or liquid interfaces with a variable surface tension, which can all induce transport and diffusion.

The understanding of active carpets may therefore prove useful in a wide range of applications. Recent advances in nano-technology enable the design of artificial cilia[63,64], origami micromachines[86,87], and four-dimensionally printed active materials[88]. One may therefore think of 'active coating materials',

for example with self-cleaning properties based on the described effect of boundary-actuated fluctuations repelling particles. One could also consider biomolecular condensates[89,90], including phase-separated biopolymers, where chemical reactions are concentrated in one phase but the products are transported to the other phase by mixing actuators at the interface. This idea of interfacial mixing may also be applied to bacteria moving on water–oil interfaces[91,92] for bioremediation of oil spills[93,94].

## Methods

**Individual flow fields.** The flow generated by an individual actuator is written in terms of the Blake tensor[74], given by

$$\mathcal{B}_{ij}(\boldsymbol{r}, \boldsymbol{r}_a) = (-\delta_{jk} + 2h\delta_{kz}(\nabla_a)_j + h^2 M_{jk}\nabla_a^2)\mathcal{J}_{ik}(\boldsymbol{r}, \boldsymbol{r}_a), \quad (12)$$

where the flow is evaluated at position $\boldsymbol{r}$ and the actuator is located at position $\boldsymbol{r}_a = (x_a, y_a, z_a = h)$ and oriented along $\boldsymbol{p}_a = (p_x, p_y, p_z)$. The indices $i, j, k \in \{x, y, z\}$ denote Cartesian components, the mirror matrix is $M_{jk} = \mathrm{diag}(1, 1, -1)$, and the Oseen tensor is

$$\mathcal{J}_{ij}(\boldsymbol{r}, \boldsymbol{r}_a) = \frac{1}{|\boldsymbol{d}|}\left(\delta_{ij} + \frac{d_i d_j}{|\boldsymbol{d}|^2}\right), \quad (13)$$

with distance $\boldsymbol{d} = \boldsymbol{r} - \boldsymbol{r}_a$. All the derivatives $\nabla_a$ of the Oseen tensor in Eq. (12) are

taken with respect to the actuator position $r_a$. As described for example in ref. [51], the individual flows can then be written as a multipole expansion of this Blake tensor.

The flow due to a point force $f_\parallel$ oriented parallel to the surface is given by

$$u_\parallel = f_\parallel \mathcal{B} \cdot p_a. \tag{14}$$

Similarly, the flow due to a point force $f_\perp$ oriented perpendicular to the surface is

$$u_\perp = f_\perp \mathcal{B} \cdot e_z. \tag{15}$$

Note, we have scaled the forces as $f = f_N/(8\pi\mu)$, where $\mu \sim 10^{-3}$ Pa s is the viscosity of water, so $f_N$ are in Newtons and $f$ have units $[m^2/s]$. The Stokes dipole flow with prefactor $\kappa$ in $[m^3/s]$ is found by taking the derivative along the $p_a$ direction,

$$u_D = \kappa(p_a \cdot \nabla_a)(\mathcal{B} \cdot p_a). \tag{16}$$

Finally, the Stokes rotlet describes an actuator that generates a torque about the $z$-axis. Its flow with prefactor $\varrho$ in $[m^3/s]$ is defined by

$$u_R = \varrho \frac{(e_x \cdot \nabla_a)(\mathcal{B} \cdot e_y) - (e_y \cdot \nabla_a)(\mathcal{B} \cdot e_x)}{2}. \tag{17}$$

These flows fields are quite complex when evaluated algebraically. Therefore, to make progress we approximate the flows as being evaluated far from the surface, such that $z_a = \epsilon z$ with $\epsilon \ll 1$. For each of the individual actuator flows ((14)–(17)) we then perform a Taylor expansion,

$$u(r, r_a, p_a) = u|_{\epsilon=0} + \frac{\partial u}{\partial \epsilon}\Big|_0 \epsilon + \frac{1}{2}\frac{\partial^2 u}{\partial \epsilon^2}\Big|_0 \epsilon^2 + \mathcal{O}(\epsilon^3), \tag{18}$$

and we only keep the leading-order contribution.

To be explicit, for the parallel Stokeslets evaluated at $r = (0, 0, z_0)$, and using $r_a = (\rho_a \cos\theta_a, \rho_a \sin\theta_a, h)$ and $p_a = (\cos\phi_a, \sin\phi_a, 0)$, this gives the Cartesian components

$$\hat{x} \cdot u_\parallel\big|_{\rho=0} = \frac{12hz_0\rho_a^2 \cos(\theta_a) \cos(\theta_a - \phi_a)}{(\rho_a^2 + z_0^2)^{5/2}} f_\parallel + \mathcal{O}(\epsilon^2), \tag{19}$$

$$\hat{y} \cdot u_\parallel\big|_{\rho=0} = \frac{12hz_0\rho_a^2 \sin(\theta_a) \cos(\theta_a - \phi_a)}{(\rho_a^2 + z_0^2)^{5/2}} f_\parallel + \mathcal{O}(\epsilon^2), \tag{20}$$

$$\hat{z} \cdot u_\parallel\big|_{\rho=0} = -\frac{12hz_0^2\rho_a \cos(\theta_a - \phi_a)}{(\rho_a^2 + z_0^2)^{5/2}} f_\parallel + \mathcal{O}(\epsilon^2). \tag{21}$$

Similarly, for the perpendicular Stokeslets with the same position and orientation $p_a = (0, 0, 1)$ we have

$$\hat{x} \cdot u_\perp|_{\rho=0} = \frac{6h^2z_0\rho_a \cos(\theta_a)(2\rho_a^2 - 3z_0^2)}{(\rho_a^2 + z_0^2)^{7/2}} f_\perp + \mathcal{O}(\epsilon^3), \tag{22}$$

$$\hat{y} \cdot u_\perp|_{\rho=0} = \frac{6h^2z_0\rho_a \sin(\theta_a)(2\rho_a^2 - 3z_0^2)}{(\rho_a^2 + z_0^2)^{7/2}} f_\perp + \mathcal{O}(\epsilon^3), \tag{23}$$

$$\hat{z} \cdot u_\perp|_{\rho=0} = \frac{6h^2z_0^2(2z_0^2 - 3\rho_a^2)}{(\rho_a^2 + z_0^2)^{7/2}} f_\perp + \mathcal{O}(\epsilon^3). \tag{24}$$

For Stokes dipoles oriented parallel to the surface, with orientation $p_a = (\cos\phi_a, \sin\phi_a, 0)$, we have

$$\hat{x} \cdot u_D|_{\rho=0} = $$
$$\frac{3hz_0\rho_a(2z_0^2(\cos(2\phi_a - \theta_a) + 3\cos(\theta_a)) - \rho_a^2(3\cos(2\phi_a - \theta_a) + 5\cos(2\phi_a - 3\theta_a) + 4\cos(\theta_a)))}{(\rho_a^2 + z_0^2)^{7/2}} \kappa + \mathcal{O}(\epsilon^2), \tag{25}$$

$$\hat{y} \cdot u_D|_{\rho=0} = $$
$$\frac{3hz_0\rho_a(\rho_a^2(-3\sin(2\phi_a - \theta_a) + 5\sin(2\phi_a - 3\theta_a) - 4\sin(\theta_a)) + 2z_0^2(\sin(2\phi_a - \theta_a) + 3\sin(\theta_a)))}{(\rho_a^2 + z_0^2)^{7/2}} \kappa + \mathcal{O}(\epsilon^2), \tag{26}$$

$$\hat{z} \cdot u_D|_{\rho=0} = \frac{6hz_0^2(\rho_a^2(5\cos(2(\phi_a - \theta_a)) + 3) - 2z_0^2)}{(\rho_a^2 + z_0^2)^{7/2}} \kappa + \mathcal{O}(\epsilon^2), \tag{27}$$

Finally, for the rotlet dipoles we have

$$\hat{x} \cdot u_R|_{\rho=0} = \frac{6hz_0\rho_a \sin(\theta_a)}{(\rho_a^2 + z_0^2)^{5/2}} \varrho + \mathcal{O}(\epsilon^3), \tag{28}$$

$$\hat{y} \cdot u_R|_{\rho=0} = -\frac{6hz_0\rho_a \cos(\theta_a)}{(\rho_a^2 + z_0^2)^{5/2}} \varrho + \mathcal{O}(\epsilon^3), \tag{29}$$

$$\hat{z} \cdot u_R|_{\rho=0} = 0, \tag{30}$$

and so forth for higher-order multipole flows.

**Characterising fluctuations: simulation details.** To characterise the strength of the active fluctuations, we simulate the distribution $\text{PDF}(v)$ of the total flow evaluated at particle position $r_0 = (0, 0, z_0)$. The carpet consists of $N_a = 4nL^2$ actuators that are randomly distributed with uniform surface density $n$ within $x, y \in [-L, L]$, with carpet size $L$ and vertical position $z_a = h$. Once the carpet configuration is sampled with standard random number generators, we evaluate the total flow $v = \sum_a u$ due to all $N_a$ actuators. We repeat this simulation to obtain an ensemble of $N_e$ total flow velocities, each due to an independent carpet realisation. Hence, we evaluate the distribution $\text{PDF}(v)$ and its moments as a function of distance from the carpet, $z_0$. In non-dimensionalised simulation units, we use the parameters $h = 1$, $n = 0.1$, $N_e = 10^4$, and $L = 500$ so $N_a = 10^5$. We have verified that $L$ is large enough to avoid edge effects. The actuator strengths used for the four types are $f_\parallel = f_\perp = \frac{3}{4}$ and $\kappa = \varrho = 30$, respectively. These simulations are presented in Fig. 1. Note the results are shown in simulation units in order to keep the description general. However, for any specific application, dimensional quantities (with standard SI units) can be inserted in all the equations throughout the paper.

**Characterising fluctuations: theory details.** We consider active carpets made of uniformly distributed actuators, so the average flow vanishes in the absence of gradients, $\langle v \rangle = 0$. This may be demonstrated by integrating any of the expressions Eqs. (19)–(30) with a constant carpet distribution, $F = n/2\pi$, which yields the average total flow

$$\langle v(r) \rangle = \int u(r, r_a, p_a) F(r_a, p_a) dr_a dp_a, \tag{31}$$

$$= \int_0^\infty \int_{-\pi}^\pi \int_{-\pi}^\pi u \frac{n}{2\pi} \rho_a d\rho_a d\theta_a d\phi_a = 0. \tag{32}$$

The variance tensor of the active fluctuations is calculated in the same way, by evaluating the integral

$$\mathcal{V}_{ij} = \langle v_i v_j \rangle = \int u_i u_j F dr_a dp_a. \tag{33}$$

To give an explicit example, the vertical component of the variance for parallel Stokeslets is given by

$$\langle v_z^2 \rangle = \int \left( \frac{12hz_0^2\rho_a \cos(\theta_a - \phi_a)}{(\rho_a^2 + z_0^2)^{5/2}} f_\parallel \right)^2 \frac{n}{2\pi} \rho_a d\rho_a d\theta_a d\phi_a \tag{34}$$

$$= \int \frac{144\pi n h^2 f_\parallel^2 z_0^4 \rho_a^2}{(\rho_a^2 + z_0^2)^5} \rho_a d\rho_a \tag{35}$$

$$= \frac{6\pi n h^2 f_\parallel^2}{z_0^2}. \tag{36}$$

This result corresponds to Eq. (2) in the main text, and is compared with simulations in Fig. 1m. We repeat this calculation for the different actuator types, and for the different components, $i, j$. Note that the off-diagonal components vanish, so the only non-trivial results are $i = j$. These results are all listed in Table 1.

Besides variance, the distributions of the active fluctuations also feature skewness

$$\langle v_i v_j v_k \rangle = \int u_i u_j u_k F dr_a dp_a, \tag{37}$$

and kurtosis,

$$\langle v_i v_j v_k v_l \rangle = \int u_i u_j u_k u_l F dr_a dp_a. \tag{38}$$

For all these quantities, we find that the only non-zero results are diagonal, where $i = j = k = l$. These results are listed in the bottom rows of Table 1 for all actuator types, and plotted in the insets of Fig. 1i–l.

**Simulating the diffusivity of particles near carpets made of fluctuating forces.** We first consider the motion of a tracer particle above a carpet of fluctuating forces, composed of $N_a$ perpendicular Stokeslets (Eq. (22)) that have a fixed position on the wall. The point forces have vertical position $h = 1$ and horizontally they are randomly distributed over the surface, with uniform density $n = 0.1$ per unit area, and within a simulation box of dimensions $x, y \in [-L, L]$ with $L = 500$ so $N_a = 4nL = 10^5$. We have verified that the simulation box is large enough to avoid edge effects by testing that the results are independent of large $L$. The tracer is initially located at $(0, 0, z_0)$. The forces do not interact with each other. Each perpendicular Stokeslet force $f_\perp = f_a(t)$ evolves dynamically in time following its own independent Ornstein–Uhlenbeck process.

This Ornstein–Uhlenbeck process is defined as

$$\frac{df}{dt} = -\frac{f}{\tau} + \sigma\eta(t), \tag{39}$$

where $\tau$ is a relaxation time, $\sigma$ is a constant that relates to the force strength, and $\eta$ is Gaussian white noise with $\langle\eta(t)\rangle = 0$ and $\langle\eta(t_1)\eta(t_2)\rangle = \delta(t_1 - t_2)$. The solution of this stochastic differential equation is

$$f(t) = f(0)e^{-t/\tau} + \sigma\int_0^t e^{-(t-t_1)/\tau}\eta(t_1)dt_1. \tag{40}$$

This corresponds to an overdamped relaxation process driven by fluctuations, which admits a Gaussian stationary distribution

$$\mathrm{PDF}\,(f) = \sqrt{\frac{1}{\pi\sigma^2\tau}}\exp\left(-\frac{f^2}{\sigma^2\tau}\right), \tag{41}$$

with mean $\langle f\rangle = 0$ and force magnitude $\langle f^2\rangle = \frac{\sigma^2\tau}{2}$. Importantly, the temporal correlation function at steady state for this Ornstein–Uhlenbeck process is also known exactly,

$$\langle f(t')f(t'')\rangle = \frac{\sigma^2\tau}{2}\exp\left(-\frac{|t'-t''|}{\tau}\right). \tag{42}$$

In our simulations, we use an average force magnitude of $\langle f^2\rangle = (3/4)^2$ and $\tau = 1/10$ so that $\sigma^2 = 2\langle f^2\rangle/\tau$, and the initial force $f(t=0)$ for each actuator is drawn randomly from the stationary distribution (Eq. (41)). Besides these active fluctuations, we do not include additional Brownian thermal fluctuations here.

The tracer's equation of motion is then given by

$$\frac{dr}{dt} = v(r(t), t) = \sum_a u(r(t), r_a, f_a(t)), \tag{43}$$

which is integrated numerically, along with the $N_a$ Ornstein–Uhlenbeck processes for all the actuators, using a fourth-order Runge–Kutta scheme. This gives one tracer trajectory. We repeat this for an ensemble of $N_e = 100$ independent trajectories, each with an independent carpet realisation composed of actuators that are distributed at different random but fixed positions, and with independent Ornstein–Uhlenbeck processes. We then repeat all this for each starting point $z_0$.

By averaging over the ensemble of $N_e$ trajectories we compute the VCFs, $C(|t'-t''|) = \langle v_i(r,t')v_j(r,t'')\rangle$, shown in Fig. 2b for different values of $z_0 \in [2, 20]$. Similarly, we determine the ensemble-averaged mean-square displacements, $\langle\Delta r_i\Delta r_j\rangle$, shown in Fig. 2c. From the latter data we also determine the components of the diffusion tensor, by statistical regression of the expression $2D_{ij}t = \langle\Delta r_i\Delta r_j\rangle$ in the regime $t \gg \tau$, the diffusive limit, which is shown in Fig. 2d.

**Derivation of the MSD and space-dependent diffusivity.** Analytical progress is made by considering small tracer displacements,

$$\langle\Delta r^2\rangle \ll r_0^2. \tag{44}$$

Then their equation of motion reduces to $\frac{dr}{dt} \approx v(r_0, t)$, plus higher-order terms that can be expanded as a power series in $1/z_0$[51]. Consequently, the MSD is given by

$$\langle\Delta r_i\Delta r_j\rangle = \left\langle\int_0^t v_i(r_0, t')dt'\int_0^t v_j(r_0, t'')dt''\right\rangle \tag{45}$$

$$= \int_0^t\int_0^t\langle v_i(r_0, t')v_j(r_0, t'')\rangle dt'dt'', \tag{46}$$

where we identify the VCF of the total flow at position $r_0$, that is

$$C_0(|t'-t''|) = \langle v_i(r_0, t')v_j(r_0, t'')\rangle. \tag{47}$$

The relationship between this ensemble-averaged flow correlation and the Ornstein–Uhlenbeck force correlations is given by

$$\frac{\langle v(t')v(t'')\rangle}{\langle v(0)v(0)\rangle} = \frac{\langle f(t')f(t'')\rangle}{\langle f(0)f(0)\rangle} = \exp\left(-\frac{|t'-t''|}{\tau}\right), \tag{48}$$

where we used Eq. (42). This expression is shown as a red dashed line in Fig. 2b. Hence, using the variance $\langle v^2\rangle = \langle v(0)v(0)\rangle$ of the active fluctuations (Eq. (33)), for example, for vertical displacements due to the perpendicular Stokeslets, the MSD simplifies to

$$\langle\Delta z^2\rangle = \frac{15\pi nh^4}{z_0^4}\frac{\sigma^2\tau}{2}\int_0^t\int_0^t\exp\left(-\frac{|t'-t''|}{\tau}\right)dt'dt'', \tag{49}$$

and similarly for other actuator types. After performing the final integrals, we obtain the MSD

$$\langle\Delta z^2\rangle = \frac{15\pi nh^4}{z_0^4}\left(t + \tau(e^{-t/\tau} - 1)\right)\sigma^2\tau^2 \tag{50}$$

$$= \begin{cases} \langle v_z^2\rangle t^2, & \text{if } t \ll \tau, \\ 2\langle v_z^2\rangle\tau t, & \text{if } t \gg \tau, \end{cases} \tag{51}$$

and similarly for the other directions $\langle\Delta r_i\Delta r_j\rangle$. This MSD transition from ballistic to diffusive motion is compared with simulations in Fig. 2c (dashed line). The corresponding anisotropic diffusivity is then given by

$$D_{xx}(z_0) = \langle v_x^2\rangle\tau = \frac{3\pi nh^4\langle f^2\rangle}{z_0^4}\tau = D_{yy}(z_0), \tag{52}$$

$$D_{zz}(z_0) = \langle v_z^2\rangle\tau = \frac{15\pi nh^4\langle f^2\rangle}{z_0^4}\tau. \tag{53}$$

These expressions are shown in Fig. 2d. The same analysis can also be applied to all other actuator types using the variances listed in Table 1.

Importantly, we should come back to the initial assumption of small displacements (Eq. (44)) and analyse its consequences. Rewriting this condition as $z^2 \gg 2D_{zz}t = 30\pi nh^4\langle f^2\rangle\tau t/z^4$ using Eq. (53), we can rearrange this for a temporal condition, $t/\tau \ll z^6/(30\pi nh^4\langle f^2\rangle\tau^2)$. We also require that $t \gg \tau$ for the particle motion to transition from ballistic to diffusive motion (see Eq. (51)). In other words, the theory is only expected to hold when the time scale of diffusive transport is slow compared to the time scale of the active fluctuations themselves. This is true far from the surface, where

$$z \gg (30\pi nh^4\langle f^2\rangle\tau^2)^{1/6}, \tag{54}$$

and similarly for other actuator types. Inserting the values used for simulations in Fig. 2a–d, being $n = 0.1$, $h = 1$, $\langle f^2\rangle = (3/4)^2$, $\tau = 0.1$, we find $z \gg 0.61$. This condition ensures that we determine the local diffusivity, $D(z)$ with small variations in $z$. Once this local diffusivity is determined from the MSDs within this limit, the global stochastic dynamics of particles leaving this local area (with large variations in $z$) can be solved using the space-dependent Fick's laws, as discussed in 'Generalised Fick's laws'.

**Simulating the diffusivity of particles near carpets made of moving actuators.** Here, we consider the motion of a tracer particle above a carpet of $N_a$ parallel Stokeslets (Eq. (19)) that each move with velocity $V = Vp_a$ along their director, $p_a$, with a constant speed $V$. They generate a flow with force $f = f_\parallel p_a$. The Stokeslets move along the surface, so $p_z = 0$, with an orientation angle $\phi_a = \arctan(p_y/p_x)$ that is subject to rotational diffusion. That is, $\frac{d\phi_a}{dt} = \sqrt{2D_r}\eta(t)$ where $\eta$ is white Gaussian noise with $\langle\eta(t)\rangle = 0$ and $\langle\eta(t_1)\eta(t_2)\rangle = \delta(t_1 - t_2)$. The parallel Stokeslets are all independent of each other. We use the parameters $V = 1$, $D_r = 10$, and $f_\parallel = 3/4$, such that $\tau_r \ll \tau_u$ for all $z_0 \in [2, 20]$. As before, the actuators are distributed randomly with uniform density $n = 0.1$ in a simulation box, $x, y \in [-L, L]$, with $L = 500$ and $h = 1$. Periodic boundary conditions are imposed so that the surface density $n = 0.1$ of the actuators remains constant. Again, besides these active fluctuations, we do not include additional Brownian thermal fluctuations here.

These $N_a + 1$ equations of motion, for the tracer and the actuators, are then integrated as before over a time period $t \in [0, 10]$ for each tracer trajectory. We repeat this simulation for an ensemble of $N_e = 100$ independent trajectories, each with an independent carpet realisation with actuators that are initially distributed at different random positions and orientations. Again, we repeat all this for ten different initial positions $z_0 \in [2, 20]$. The ensemble-averaged results are shown in Fig. 2e–h.

**Generalised Fick's laws.** For clarity, we first revise the case of constant diffusion $\tilde{D}$ in one spatial dimension, $z$. Then, Fick's first law relates gradients in concentration $\varphi(z, t)$ and an external drift $v_d$ to the total flux,

$$J_z = v_d\varphi - \tilde{D}\partial_z\varphi. \tag{55}$$

Together with the continuity equation, $\partial_t\varphi = -\partial_z J_z$, this gives Fick's second law,

$$\frac{\partial\varphi}{\partial t} = -\frac{\partial}{\partial z}(v_d\varphi) + \tilde{D}\frac{\partial^2\varphi}{\partial z^2}, \tag{56}$$

which is also known as the Fokker–Planck equation. This is equivalent to motion described by the following Langevin equation, $\frac{dz}{dt} = v_d + \sqrt{2\tilde{D}}\eta(t)$, where $\eta$ is Gaussian white noise as defined below Eq. (39).

Rather than being constant in space, the diffusivity of tracers near an active carpet depends continuously on position. Such stochastic processes can often be described by an effective Smoluchowski equation[78], rather than standard Langevin methods which make no reference to individual collisions. Here, we follow this approach as a foundation for the generalised Fick's laws that describe the diffusion of a tracer particle as a function of distance from an active carpet. Since we only have gradients in the vertical direction, we use the short-hand notations $D(z) = D_{zz}$ for the vertical diffusivity and $v(z) = \sqrt{\langle v_z^2\rangle} = \sqrt{\mathcal{V}_{zz}}$ for the mean vertical speed.

With this spatial dependence, the question arises whether the second term in Eq. (55) should be interpreted as $D\partial_z\varphi$ or $\partial_z(D\varphi)$, or something in between. This question is not well posed, because it depends on the physical processes in question. In other words, generalising this expression for macroscopic quantities requires partial knowledge of the microscopic mechanism for diffusion. In particular, information is needed about the spatial dependence of the memory time $\tau(z)$, and of the mean vertical speed, $v(z)$. The diffusivity can then be written as the combination of these ingredients, $D(z) = v^2(z)\tau(z)$, as shown in Eq. (53). Indeed,

either a larger average speed or a longer time between reorientations can give rise to a larger diffusivity. Note that for Ornstein–Uhlenbeck forcing the time scale $\tau$ does not depend on position, but this need not be true in general. For example, for rapidly moving actuators the smallest decorrelation time is $\tau \sim z/V$. On the other hand, for slowly moving actuators the rotational diffusion constant $D_r$ sets the smallest memory time.

Using this information about the microscopic interactions, a 'telegraph model'[78] can be constructed that describes the space-dependent diffusion process. Inspired by this model, we consider two particle populations, with population densities $\varphi_{\text{up}}(z,t)$ and $\varphi_{\text{down}}(z,t)$, respectively, that either move up or down along $z$ with mean speed $v(z)$ due to the active carpet fluctuations. The mean speed is the same for the two populations at any given $z$ since we consider a uniform carpet without net drift, $\langle v \rangle = 0$, as shown in Eq. (32). The particles switch directions at a mean rate of $\frac{1}{2}\tau^{-1}$, set by the memory time of the active carpet. The total particle density is then given by $\varphi(z,t) = \varphi_{\text{up}} + \varphi_{\text{down}}$, and the diffusive flux of particles is $J_{\text{diff}}(z,t) = v(\varphi_{\text{up}} - \varphi_{\text{down}})$. Subsequently, using continuity of particle flow and conservation of particle number, the up and down populations evolve according to

$$\frac{\partial \varphi_{\text{up}}}{\partial t} = -\frac{\partial(v\varphi_{\text{up}})}{\partial z} - \frac{\varphi_{\text{up}}}{2\tau} + \frac{\varphi_{\text{down}}}{2\tau}, \tag{57}$$

$$\frac{\partial \varphi_{\text{down}}}{\partial t} = \frac{\partial(v\varphi_{\text{down}})}{\partial z} + \frac{\varphi_{\text{up}}}{2\tau} - \frac{\varphi_{\text{down}}}{2\tau}. \tag{58}$$

In each equation, the first term describes spatial gradients in the moving populations, while the last two terms describe the switching between particles moving up and down, and vice versa. By adding and subtracting, the equations (57) can be rewritten in terms of the total density and diffusive flux, giving

$$\frac{\partial \varphi}{\partial t} = -\frac{\partial J_{\text{diff}}}{\partial z}, \tag{59}$$

$$\frac{\partial(J_{\text{diff}}/v)}{\partial t} = -\frac{\partial(v\varphi)}{\partial z} - \frac{J_{\text{diff}}}{v\tau}. \tag{60}$$

Taking the time derivative of the first expression and combining with the second expression yields

$$\frac{\partial^2 \varphi}{\partial t^2} = \frac{\partial}{\partial z}\left(v\frac{\partial(v\varphi)}{\partial z}\right) + \frac{\partial}{\partial z}\left(\frac{J_{\text{diff}}}{\tau}\right). \tag{61}$$

Then, we assume that the high-frequency behaviour of particle movements can be neglected, so the second time derivative on the left-hand side vanishes,

$$0 = \frac{\partial}{\partial z}\left(v\frac{\partial(v\varphi)}{\partial z} + \frac{J_{\text{diff}}}{\tau}\right). \tag{62}$$

Integrating this expression, we can solve for the diffusive flux $J_{\text{diff}}$. The constant of integration is set equal to zero to ensure that $J_{\text{diff}}$ vanishes when the variance of the active fluctuations $v^2$ are zero. If the particles are sedimenting with a constant drift velocity $v_d$, there is an additional flux $J_{\text{drift}} = v_d\varphi$, so the total flux is $J_z = J_{\text{diff}} + J_{\text{drift}}$. Then, combining all this information gives the first generalised Fick's law in the vertical direction,

$$J_z = v_d\varphi - v^2\tau\frac{\partial \varphi}{\partial z} - v\varphi\tau\frac{\partial v}{\partial z} \tag{63}$$

$$= v_d\varphi - \mathcal{V}_{zz}\tau\frac{\partial \varphi}{\partial z} - \frac{\varphi\tau}{2}\frac{\partial \mathcal{V}_{zz}}{\partial z} \tag{64}$$

We find the other components of the flux by repeating the telegraph model analysis in the $x$ and $y$ directions, which gives

$$J_x = v_x^d\varphi - \mathcal{V}_{xx}\tau\frac{\partial \varphi}{\partial x} - \frac{\varphi\tau}{2}\frac{\partial \mathcal{V}_{xx}}{\partial x}, \tag{65}$$

$$J_y = v_y^d\varphi - \mathcal{V}_{yy}\tau\frac{\partial \varphi}{\partial y} - \frac{\varphi\tau}{2}\frac{\partial \mathcal{V}_{yy}}{\partial y}. \tag{66}$$

Since the variance tensor only has diagonal components, we can write the generalised flux in three dimensions as

$$\boldsymbol{J}(\boldsymbol{r}) = \boldsymbol{v}_d\varphi - (\tau\mathcal{V}\cdot\nabla)\varphi - \frac{\varphi\tau}{2}\nabla\cdot\mathcal{V}, \tag{67}$$

which is written in the main text as Eq. (6). Note that the last term only has a vertical component for systems that obey translational invariance along the horizontal directions, when the variance tensor only depends on $z$. Finally, using the continuity equation we obtain the second generalised Fick's law,

$$\frac{\partial \phi}{\partial t} = -\partial_i(v_i^d\varphi) + \partial_i(\tau\mathcal{V}_{ij}\partial_j\varphi) + \partial_i\left(\frac{\varphi\tau}{2}\partial_j\mathcal{V}_{ij}\right), \tag{68}$$

where repeated indices are summed over.

### Sedimentation towards an active carpet: simulation details.

Here, we consider the motion of a sedimenting tracer particle above a carpet of moving parallel Stokeslets. For spherical particles, the sedimentation is described by a constant drift velocity $\boldsymbol{v}_d = -v_g\hat{z} = \frac{d^2\Delta\rho}{18\mu}\boldsymbol{g}$, in terms of the gravitational acceleration $\boldsymbol{g}$, the

particle diameter $d$, its density difference with the medium $\Delta\rho$, and the medium viscosity $\mu$. The equations of motion of the moving forces are as described in 'Methods: Simulating the diffusivity of particles near carpets made of moving actuators', with the parameters $V = 1$, $D_r = 10$, $n = 0.1$, $L = 500$, $h = 1$ and $f_{\parallel} = 10$. For the tracer equation of motion we add the sedimentation, and we introduce a reflecting boundary condition at $z = h$ to prevent the particles from crossing the active carpet. This system is integrated numerically for different sedimentation velocities, $v_g \in [10^{-2}, 1]$. We simulate over a long period of time, $t \in [0, 10^5]$, to ensure that the sedimentation profile is well sampled. To clarify, we do not average this sedimentation profile over a statistical ensemble of independent carpet configurations. Therefore, since the only averaging is temporal, the results are informative about the dynamics of a given system. We then normalise this particle concentration profile,

$$\Phi(z) = \text{PDF}(z) = \varphi(z)/N_p, \tag{69}$$

where $N_p = \int\varphi dz$ is the number of tracers in the system, so $\int\Phi(z)dz = 1$. From these distributions we also evaluate the maximum $z_{\text{max}}$ where $\frac{d\varphi}{dz} = 0$. These results are shown and compared with our analytical predictions in Fig. 3.

### Self-cleaning effect.

These simulation results can be understood using the generalised Fick's laws we discussed earlier. The first two terms on the RHS in Eq. (63) are identical to the case of constant diffusion (Eq. (55)), describing a flux of particles towards regions of low concentration. A third emerges, however, which describes diffusion towards regions of low fluid speed. Since the active fluctuations decay with distance (e.g. see Eq. (2)), this term leads to a flux directed away from the carpet.

To understand this better, we must quantify the contributions of the flux. Since the variance of all the active fluctuations in Table 1 feature an algebraic decay, we write

$$\mathcal{V}(z) = \tilde{\mathcal{V}}/z^{\alpha}. \tag{70}$$

Similarly, for the memory time we write

$$\tau(z) = \tilde{\tau}/z^{\beta}, \tag{71}$$

because this algebraic form is common in natural systems. To name a few examples: $\beta = 0$ corresponds to a constant memory time. $\beta = -1$ corresponds to the time scale $\tau \sim z/v_a$ associated with an actuator moving underneath a tracer particle. $\beta = -2$ corresponds to the time scale $\tau \sim z^2/D_a$ associated with actuators diffusing underneath a tracer particle. Note that one could have inverted the exponent to keep $\beta$ positive, but this does not change anything physically. Hence, we prefer to keep the expressions for $\alpha$ and $\beta$ consistent.

Then the vertical diffusivity is $D(z) = \tilde{D}/z^{\alpha+\beta}$, where the constant part is $\tilde{D} = \tilde{\mathcal{V}}_{zz}\tilde{\tau}$. For example, for slowly moving parallel Stokeslets we have $\tilde{\mathcal{V}}_{zz} = 6\pi n h^2 f_{\parallel}^2$ and $\tilde{\tau} = D_r^{-1}$ with $\alpha = 2$ and $\beta = 0$ from Eq. (5). Inserting the expressions (70)–(71) into (63), one obtains

$$J_z(z) = v_g\varphi - \frac{\tilde{D}}{z^{\alpha+\beta}}\frac{\partial \varphi}{\partial z} + \frac{\alpha\tilde{D}}{2z^{\alpha+\beta+1}}\varphi. \tag{72}$$

The first term is negative for sedimentation, and the second term still describes ordinary diffusion towards regions of low concentration. But the third term is always positive, repelling particles away from the carpet, which explains the self-cleaning effect.

To solve the sedimentation profile, we require that $J_z = 0$ at steady state, which yields

$$\frac{\varphi(z)}{\varphi_0} = z^{\alpha/2}\exp\left(-\frac{v_g z^{\alpha+\beta+1}}{\tilde{D}(\alpha+\beta+1)}\right), \tag{73}$$

where $\varphi_0$ is a normalisation factor. In the limit of a constant diffusivity ($\alpha = \beta = 0$) we recover the Boltzmann distribution, $\varphi(z) = \varphi_0 e^{-v_g z/\tilde{D}}$. This is no longer true for active carpets with decaying fluctuations because of the $z^{\alpha/2}$ factor, where $\alpha = 2$ for parallel Stokeslets. Therefore, the sedimentation profile features a maximum (Fig. 3c), which is located at

$$z_{\text{max}} = \left(\alpha\tilde{D}/2v_g\right)^{1/(\alpha+\beta+1)}. \tag{74}$$

These results agree with our simulations (Fig. 3c, d), for different values of the sedimentation velocity.

### Sedimentation with active and thermal diffusion.

Besides the fluctuating flows generated by the active carpet, the particles may also experience Brownian thermal fluctuations. This thermal diffusion $D_{\text{th}}$ can be included explicitly in the generalised Fick's law:

$$J_z(z) = -v_g\varphi - D_{\text{th}}\frac{\partial \varphi}{\partial z} - \frac{\tilde{D}}{z^{\alpha+\beta}}\frac{\partial \varphi}{\partial z} + \frac{\alpha\tilde{D}}{2z^{\alpha+\beta+1}}\varphi. \tag{75}$$

Then, the expression $J_z = 0$ can still be solved analytically to determine the steady-state sedimentation profile. For parallel Stokeslets, for example, with $\alpha = 2$

and $\beta = 0$, we find the solution

$$\frac{\varphi(z)}{\varphi_0} = \frac{z}{\sqrt{\tilde{D} + D_{th}z^2}} \exp\left(\sqrt{\frac{\tilde{D}}{D_{th}^3}} \nu_g \tan^{-1}\left(\frac{\sqrt{D_{th}}z}{\sqrt{\tilde{D}}}\right) - \frac{\nu_g z}{D_{th}}\right). \tag{76}$$

It is important to note that this function has the same shape as the original solution (Eq. (73)). Indeed, particles are still repelled from the active surface, $\lim_{z\to 0}\varphi(z) = 0$, and the function has a maximum at the same location as before, at $z_{max} = (\tilde{D}/\nu_g)^{1/3}$ for all $D_{th} \geq 0$. Thus, the self-cleaning effect is not affected by thermal diffusion. Of course, when the surface activity vanishes, $\tilde{D} \to 0$, we recover the Boltzmann distribution.

**Diffusion from a source to an active carpet sink: simulation details.** In this section, we consider the dynamics of particles that are spawned at a source and absorbed by an active sink. The particles are subject to active fluctuations due to a carpet of slowly moving parallel Stokeslets. The equations of motion of the $N_a$ actuators are as described in 'Methods: Simulating the diffusivity of particles near carpets made of moving actuators and Sedimentation towards an active carpet: simulation details'. For the tracer equation of motion we remove the sedimentation, we impose an absorbing boundary condition at $z_{sink} = h$, and a reflecting boundary condition at $z_{source} = H$. Whenever a particle is absorbed by the sink, we place it back at the source, at $x = y = 0$, and we redistribute all the actuators with new random positions and orientations to start a new fully independent trajectory. We run two separate types of simulations: First, the gap size is varied with different values of $H \in [2, 20]$ with constant force $f_{\parallel} = 10$. Second, we vary $f_{\parallel} = \in [1, 10]$ with constant source height $H = 5$. For each of these forces we measure the flux $J_z$, defined as the number of particles that diffuse from the source to the sink per unit time, $J_z = -N_p/\langle t_{mfp}\rangle$, where $\langle t_{mfp}\rangle$ is the mean first-passage time and $N_p$ is the number of particles. By symmetry, we expect the nutrient flux to scale quadratically with the force, because the diffusion equations are invariant under the transformation $f \to -f$. Indeed, this is also observed in the simulations, as shown in Fig. 4.

**Diffusion from a source to an active carpet sink: theory details.** To solve the system of non-equilibrium diffusion from a source to an active sink, we again consider the vertical flux given by Eq. (7). This time, the sedimentation velocity is equal to zero and we seek the steady-state solution ($\partial_t\varphi = 0$) with fixed particle concentrations at the source and the sink. Hence, we must solve the continuity equation $\partial_z J_z = 0$ subject to the boundary conditions $\varphi(H) = \varphi_+$ and $\varphi(h) = 0$. This gives the solution

$$\frac{\varphi(z)}{\varphi_+} = \frac{z^{\alpha+\beta+1} - h^{\beta+1}(hz)^{\alpha/2}}{H^{\alpha+\beta+1} - h^{\beta+1}(hH)^{\alpha/2}}, \tag{77}$$

for $\alpha \geq 0$ and $\beta \geq -1$, or a slightly more complex function for other values. The corresponding solution for the flux, equivalent to the particle capture rate, is

$$\frac{J_z}{\varphi_+} = -\frac{(\alpha + 2\beta + 2)\tilde{D}}{2\left(H^{\alpha+\beta+1} - h^{\beta+1}(hH)^{\alpha/2}\right)}. \tag{78}$$

When comparing this prediction with the simulated flux, care should be taken to account for the normalisation (Eq. (69)), because the number of particles $N_p$ is coupled to concentration $\varphi_+$ at the top boundary condition $\varphi(H) = \varphi_+$. This relationship can also be computed exactly,

$$\frac{N_p}{\varphi_+} = \int \frac{\varphi(z)}{\varphi_+} dz = \int_h^H \frac{z^{\alpha+\beta+1} - h^{\beta+1}(hz)^{\alpha/2}}{H^{\alpha+\beta+1} - h^{\beta+1}(hH)^{\alpha/2}} dz, \tag{79}$$

which depends on $H$, so the power-law of $J_z(H)$ should be rescaled based on whether $N_p$ or $\varphi_+$ is kept constant. In the limit $h \ll H$, this simplifies to

$$\frac{N_p}{\varphi_+} \approx \int_0^H \frac{z^{\alpha+\beta+1}}{H^{\alpha+\beta+1}} dz = \frac{H}{\alpha + \beta + 2}. \tag{80}$$

Hence, using Eq. (78) for $h \ll H$ yields

$$J_z(H) \approx -\varphi_+ \frac{(\alpha + 2\beta + 2)\tilde{D}}{2H^{\alpha+\beta+1}}, \tag{81}$$

$$\approx -N_p \frac{(\alpha + \beta + 2)(\alpha + 2\beta + 2)\tilde{D}}{2H^{\alpha+\beta+2}}. \tag{82}$$

For slowly moving parallel Stokeslets ($\alpha = 2$ and $\beta = 0$), we then have $J_z \propto z^{-3}$ for a constant $\varphi_+$ concentration, or $J_z \propto z^{-4}$ for a constant number of particles $N_p$. In Fig. 4c we show the latter.

**Diffusion from a source to an active carpet sink: comparison with thermal diffusion.** We expect that the thermal diffusion will be more effective at transporting the particles if the distance between the source and the sink is large, because the thermal noise does not decay with $z$. To quantify this, we equate the

active carpet flux (Eq. (81)) with the thermal flux,

$$-\varphi_+ \frac{(\alpha + 2\beta + 2)\tilde{D}}{2H^{\alpha+\beta+1}} = -\varphi_+ \frac{D_{th}}{H}. \tag{83}$$

The boundary concentration $\varphi_+$ cancels out, so we find that the value of $H$ for which the two are equal is

$$H^* = \left(\frac{(\alpha + 2\beta + 2)\tilde{D}}{2D_{th}}\right)^{1/(\alpha+\beta)}. \tag{84}$$

Inserting $\tilde{D} = 6\pi n h^2 f_{\parallel}^2/D_r$ from Eq. (5) with $\alpha = 2$ and $\beta = 0$ for parallel Stokeslets, and using the typical values $h \sim 1\,\mu m$, $n \sim 1\,\mu m^{-2}$, $D_r \sim 1\,s^{-1}$ and $8\pi\mu f_{\parallel} \sim 1$ pN, we find $H^* \sim 350$ and $11\,\mu m$, respectively, for micron-sized and molecular paxrticles of $D_{th} \sim 0.5$ and $500\mu m^2/s$.

The diffusive flux can also be computed in the presence of both thermal and active fluctuations. As before, we solve $\partial_z J_z = 0$ using the generalised Fick's law that includes thermal diffusion (Eq. (75)) without gravity, with boundary conditions $\varphi(0) = 0$ and $\varphi(H) = \varphi_+$. For parallel Stokeslets this yields the concentration profile

$$\frac{\varphi(z)}{\varphi_+} = \frac{z\left(\sqrt{\tilde{D} + D_{th}H^2} - \sqrt{\frac{\tilde{D}(\tilde{D} + D_{th}H^2)}{\tilde{D} + D_{th}z^2}}\right)}{H\left(\sqrt{\tilde{D} + D_{th}H^2} - \sqrt{\tilde{D}}\right)}, \tag{85}$$

and the corresponding flux

$$\frac{J_z}{\varphi_+} = -\left(\sqrt{\tilde{D}^2 + D_{th}\tilde{D}H^2} + \tilde{D} + D_{th}H^2\right)H^{-3}. \tag{86}$$

As expected, in the limit $\tilde{D} \to 0$ we recover the thermal flux, $J_z = -D_{th}\varphi_+/H$. This corresponds to ~50 particles/second for molecular diffusion with $D_{th} \sim 500$ $\mu m^2/s$, and using $\varphi_+ \sim 1$ particle/$\mu m$ and $H \sim 10\,\mu m$. Conversely, in the limit $D_{th} \to 0$ we recover the original solution, $J_z = -2\tilde{D}\varphi_+/H^3$. This gives a 'bare' active flux of ~60 particles/second when inserting the same values as those below Eq. (84). Interestingly, these fluxes do not just add up because there is also a cross term. In fact, the total flux from Eq. (86) gives $J_z \sim 128$ particles/second. Therefore, the thermal diffusion can actually enhance the active diffusive flux, and vice versa, since they co-operate.

**Advective and diffusive transport.** To investigate the relative importance of local advective and diffusive transport, we consider a carpet composed of perpendicular Stokeslets that fluctuate about a non-zero mean. Then, the Ornstein–Uhlenbeck process (Eq. (39)) becomes

$$\frac{df}{dt} = -\frac{f - \bar{f}}{\tau} + \sigma\eta(t), \tag{87}$$

where the mean force is $\langle f\rangle = \bar{f}$ and its variance is $\text{Var}(f) = \sigma^2\tau/2$ as before. The resulting flow is then described by an advective contribution, $\boldsymbol{v}_{adv}$ due to the mean force, $\bar{f}$, and a diffusive contribution, $\boldsymbol{v}_{diff}$ due to its variance, $\text{Var}(f)$. The mean of the diffusive contribution vanishes when averaging over the temporal noise but, at any one location, the advection does not.

Naturally, the advection is not significant in the limit of a small mean force, when $\bar{f}^2 \ll \text{Var}(f)$. Even when the mean force is comparatively large, however, the active diffusion can still dominate far from the surface, depending on the structure of $F(\boldsymbol{r}_a, \boldsymbol{p}_a)$. This is explained in terms of the local heterogeneities becoming less important when $z \gg r_{nn}$, where the typical nearest-neighbour distance between actuators is $r_{nn} \sim 1/\sqrt{n}$. To quantify this carefully, we consider a square lattice of perpendicular Stokeslets with lattice spacing $r_{nn} = \lambda$. That is, the forces are located at position $(i\lambda, j\lambda, h)$ where $i$ and $j$ are integer numbers, so the number density $n = 1/\lambda^2$. The total advection generated by this active carpet is given by

$$\boldsymbol{v}_{adv}(x, y, z) = \sum_{i=-\infty}^{\infty}\sum_{j=-\infty}^{\infty} \boldsymbol{u}_{\perp}(x - i\lambda, y - j\lambda, z, h, \bar{f}), \tag{88}$$

where $\boldsymbol{u}_{\perp}$ is given by Eq. (15). This total flow is shown in Fig. 5 for different lattice spacings, where all Stokeslets have the same (negative) force $\bar{f}$. In all cases, there is a down-welling region (downward flow) near the Stokeslets and, by incompressibility, up-welling regions between the Stokeslets. Perhaps counter-intuitively, at a given distance $z$ from the surface, the sparse carpets (Fig. 5a, with large $\lambda$) drive stronger flows than the dense carpets (Fig. 5b, with small $\lambda$). This is highlighted in Fig. 5c, which plots the vertical flow velocity along the line $y = 0$ for different values of $\lambda$. These curves show the down-welling regions around $x = 0$, $\pm\lambda$, and up-welling regions around $x = \pm\lambda/2, \pm3\lambda/2, \dots$, but their amplitude decreases strongly with decreasing $\lambda$, i.e. with increasing number density $n$. This is quantified further in Fig. 5d, which shows the vertical flow directly above a Stokeslet ($x = y = 0$). Using Eq. (22), we write the normalised total vertical flow as

$$\Phi(\zeta) = \frac{v_{adv,z}}{u_{\perp,z}}\bigg|_0 = \sum_{i,j=-\infty}^{\infty} \frac{2 - 3(i^2 + j^2)\zeta^{-2}}{2\left(1 + (i^2 + i^2)\zeta^{-2}\right)^{7/2}}, \tag{89}$$

where the dimensionless number $\zeta = z/\lambda = z\sqrt{n}$ and the normalisation factor is

$u_\perp^z(0,0,z) = 12h^2\bar{f}/z^3$. Recall that $\bar{f}$ has units m²/s because forces are scaled with the fluid viscosity (see text under Eq. (15)). Then, in the limit $\zeta \to 0$ we recover the flow due to a single Stokeslet, $\Phi \to 1$, as expected. However, in the limit $\zeta \to \infty$ the flow tends to zero because the spatial gradients in the actuator density disappear. This decay is quite strong (Fig. 5d; black points), approximately like a Gaussian function, $\Phi(\zeta) \approx \exp(-\zeta^2)$ (dashed blue line). Thus, the normalised advective transport decays rapidly with $\zeta$, while the diffusive transport actually increases. Specifically, using $\langle v_{\mathrm{diff},z}^2 \rangle = 15\pi n h^4\, \mathrm{Var}\,(f)/z^4$, we can write the normalised diffusive transport as

$$\frac{\sqrt{\langle v_{\mathrm{diff},z}^2 \rangle}}{|u_{\perp,z}|} = \zeta \sqrt{\frac{15\pi}{48}\,\frac{\mathrm{Var}\,(f)}{\bar{f}^2}}, \tag{90}$$

which is shown in Fig. 5d as red lines. The relative importance of the diffusive and the advective transport is

$$\frac{\sqrt{\langle v_{\mathrm{diff},z}^2 \rangle}}{|v_{\mathrm{adv},z}|} = \frac{\zeta}{\Phi(\zeta)} \sqrt{\frac{15\pi}{48}\,\frac{\mathrm{Var}\,(f)}{\bar{f}^2}}. \tag{91}$$

Hence, the diffusion dominates over advection beyond a distance $z^* = \zeta^*/\sqrt{n}$ from the carpet, where

$$\zeta^* = \sqrt{\frac{1}{2} W_0\left(\frac{32\bar{f}^2}{5\pi\,\mathrm{Var}\,(f)}\right)}, \tag{92}$$

in terms of the Lambert $W_0$ function. This occurs at $\zeta^* \approx 0.85$ for $\mathrm{Var}\,(f) = \bar{f}^2$, which is fairly close to the active carpet. Even when their fluctuations are a thousand times weaker (Fig. 5d; red dotted line), for $\bar{f}^2 = 10^6\,\mathrm{Var}\,(f)$, the transition occurs at $\zeta^* \approx 2.55$, which is still not that far from the carpet. This is especially relevant for high actuator densities. Indeed, many organisms like *Vorticella* colonies can grow fairly dense, approaching close packing.

Another point to note is that the advective flows can average out in time: Consider a particle located in a down-welling region, slowly moving down towards an actuator. If the particle is also subject to active and/or passive fluctuations, it can diffuse horizontally into an up-welling region, so it can escape. To demonstrate this, we repeat our diffusion simulations (cf. Fig. 2a–d) for a carpet of perpendicular Stokeslets with very weak active fluctuations, $\mathrm{Var}(f) = 10^{-6}$, compared to a strong mean force directed towards the carpet, $\bar{f} = -1$. As shown in Fig. 5e, the MSD still transitions to diffusive motion when $t > \tau$, and the ballistic advective motion disappears over time. Moreover, the resulting space-dependent diffusivity (Fig. 5f) still agrees with the theoretical prediction (Eq. (4)) for $z \gtrsim 3$ for all components of $D_{ij}$.

The nutrient flux that individual organisms receive therefore depends strongly on the distance of the nutrient source. If the source is located at $H < z^*$, then the flux to individual organisms can be large due to advection. If the source is located at $H > z^*$, then the flux is determined by diffusion. This spreads out the nutrients horizontally before they reach the surface, so on average all organisms receive the same global diffusive flux as discussed in 'Methods: Diffusion from a source to an active carpet sink: simulation details'.

The theory can also be extended for situations where the relation $\langle v_{\mathrm{diff}}^2 \rangle \gg v_{\mathrm{adv}}^2$ does not hold. This may be important for scenarios in biology or synthetic carpets of intermediate actuator densities. As a first approximation, one could explicitly insert the advection flow as $\boldsymbol{v}_d = \boldsymbol{v}_{\mathrm{adv}}(\boldsymbol{r})$ into the three-dimensional generalised flux (Eq. (67)), assuming that the fluctuations are still uniformly distributed on the surface. This advection term could be written in terms of Stokeslets, or found with any other hydrodynamic technique such as the boundary-element method, a squirmer-like model, or computational fluid dynamics (CFD) simulations. The disadvantage of this formulation is that it is inherently system specific. The advantage is that any flow pattern of interest can be inserted (e.g. ciliary transport, filter feeding, bacteria on surfaces), so the resulting advection-diffusion equations can be solved accordingly.

**Quenched disorder**. The connection between advective and diffusive transport is also related to quenched disorder, the notion that spatial heterogeneity can be frozen in place so a spatial average would not be equal to a local temporal average. In other words, a system features quenched disorder if it has random variables that are quenched (frozen) in time, so their dynamics cannot evolve as fast as the other time scales in the system. In general, active carpets can indeed feature quenched disorder. In that case, it would not be appropriate to model the tracer dynamics with the generalised Fick's laws described here: This modelling approach is based on averaging with respect to a large ensemble of active carpet configurations, so it would not always be informative about the dynamics of a single specific system configuration.

However, the disorder need not necessarily be quenched for active carpets. The relevant random variables are the relative positions and orientations between the actuators and the tracer particles. Therefore, there is no quenched disorder when the actuators meander along the surface, like bacteria, as long as they move or turn rapidly. Similarly, even if the actuators themselves are fixed, the tracers may still diffuse in space under the right conditions, so the relative positions could still vary freely. They key question is what these conditions are.

The first requirement is that the advective transport (due to individual actuators locally) is much weaker than the diffusive transport (due all actuators together, possibly aided by thermal noise). If a tracer is caught in a local actuator current, then its dynamics are effectively quenched; however, if the particle can diffuse away and escape, the ballistic motion disappears and the advection flows tend to average out over time (Fig. 5e, f). Hence, as the particles explore space horizontally, their spreading over time becomes equivalent to spatial averaging, so the dynamics become annealed. As described in previous section, the diffusion dominates advection if the particles are located far away from the active surface (see Eq. (91)).

The second requirement is that the time scale of diffusive transport is slow compared to the time scale of the active fluctuations themselves. This ensures that the tracer motion is diffusive over time and not ballistic according to a specific carpet configuration. As described in 'Methods: Derivation of the mean-squared displacement and space-dependent diffusivity', this requirement is satisfied when Eq. (54) holds. Therefore, both requirements are fulfilled beyond a certain distance from the surface.

To verify the generalised Fick's laws, we compared their predictions with detailed hydrodynamic simulations. These fully resolve all the actuator positions and orientations throughout time, so any quenched disorder is explicitly included. Importantly, all our main findings (enhanced diffusivities, sedimentation profiles, nutrient fluxes) are supported by this data. Indeed, we found that the simulations and the theory agree well with one another beyond a certain distance from the surface, when both requirements described above are fulfilled. Future theories could perhaps relax these conditions by taking the effects of quenched disorder into account. We expect this could be an exciting opportunity of further research in the field of non-equilibrium statistical mechanics and active matter systems.

**Extension to more complex geometries**. In principle, our theory may be generalised for active carpets of more complex geometries by taking the following steps: First, one should find the hydrodynamic Green's function (cf. the Blake tensor in Eq. (12)) that satisfies the Stokes equations and the boundary conditions of the geometry in question. Once this flow solution is known, one can start developing simulations to verify the following steps. Second, the mean flow $\langle \boldsymbol{v}(\boldsymbol{r}) \rangle$ should be determined by integrating this Green's function over the carpet along with its force distribution, as in Eq. (32). This may tend to zero for homogeneously distributed carpets, depending on the surface shape and the distribution of actuator positions and orientations, but not necessarily. Third, one should determine the variance tensor $\mathcal{V}_{ij}(\boldsymbol{r}) = \langle v_i v_j \rangle$ as in Eq. (33), which may in general be dependent on all three spatial coordinates. Fourth, the generalised flux may be extended by revisiting the telegraph model, as in 'Methods: Generalised Fick's laws'. These equations may then be solved numerically or analytically, but care should be taken that the conditions for the theory to be accurate are correctly translated to the new geometry.

## Data availability
All simulation data used for this paper are available from the corresponding authors upon request.

## Code availability
All simulation codes used for this paper are available from the corresponding authors upon request.

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

## Acknowledgements

F.G.-L. acknowledges Millennium Nucleus Physics of Active Matter of ANID (Chile). H. L. acknowledges support from the Deutsche Forschungsgemeinschaft, DFG projects SPP 1726 and LO 418/23. A.J.T.M. acknowledges funding from the Human Frontier Science Program (Fellowship LT001670/2017) and the United States Department of Agriculture (USDA-NIFA AFRI grants 2020-67017-30776 and 2020-67015-32330). F.G.-L. and A.M. also acknowledge support from the American Physical Society (APS) for an International Research Travel Award (IRTAP).

## Author contributions

F.G.-L. and A.J.T.M. contributed equally to this work and are joint first, last and corresponding authors. F.G.-L., H.L., and A.J.T.M. designed the research and wrote the manuscript. F.G.-L. and A.J.T.M. performed the simulations. A.J.T.M. developed the theory.

## Competing interests

The authors declare no competing interests.
