## [Peer Review File · Nature Communications]

REVIEWER COMMENTS

Reviewer #1 (Remarks to the Author):

The paper studies the transport properties of so-called active carpets, collections of agents above a no-slip surface, that stir the surrounding fluid. Active carpets generate stochastic flows in the fluid, and this can affect how suspended particulates are transported. The authors study a simplified version of active carpets, where active agents at low Reynolds number are represented in their far field limit by suitable multipoles derived from the Blake tensor at a distance "h" from the boundary. The active agents can be either moving randomly along the surface, with a given rotational diffusivity, or are fixed and exert a random perpendicular force which follows an Ornstein-Uhlenbeck process of zero mean.

The authors evaluate the statistical properties of the local velocity PDF using an approximation of the flow fields for distances large compared to "h", and use these to derive a Fokker-Planck equation for the ensuing transport of passive tracers. They then analyse two test cases, particle sedimentation and diffusion to an active sink, which show clear differences from the thermal equilibrium case. The former shows a clearance region just above the active carpet layer, while the latter can show an increase in the absorbed flux if the source is sufficiently close to the active carpet.

Overall the paper is well written, and the results presented are of high quality (but see a few points raised below). The approach is simple and elegant, and easy to follow. Active matter -both alive and synthetic- has been at the centre of a large research effort over the last years. The current paper is an excellent contribution to the general question of how activity impacts on transport at the micro-scale. Therefore, I am happy to recommend the paper for publication, once the authors address satisfactorily the points raised below.

%%%%%%%%%

%%%%%%%%%

%%%%%%%%%

1) The modelling approach is based on local average properties of the stochastic flows generated by the active agents. The average here is intended to be with respect to configurations of the active carpet.

For agents that meander along the surface, like bacteria, this is justified as long as the timescales involved in the processes (e.g. nutrient absorption) are long compared to the autocorrelation time of the swimmers' configuration. This, however, cannot be the case for fixed spatial configuration of the active carpet. In this case, the spatial heterogeneity in the flow field can be frozen in place, and a spatial average would not be equal to a local temporal average. As a result, some parts of the carpet should get a much higher flux of tracers than others. I believe that this can cause a very significant difference between a hypothetical experiment and the theory proposed in this paper.

Let me try to be more explicit. Let us consider for example the case of a carpet of Vorticella, which seems to me to be the experimental system that the authors have in mind when dealing with perpendicular Stokeslet. In this case: i) the cells are fixed in position; ii) the point force that they exert can change in magnitude but is always towards the wall. In this case, although the space averaged flow is still zero, the individual cells should get a massive increase in nutrient uptake due to direct advection towards their mouth. In the end, it is this flux of nutrients that is important for the organism, not the global flux towards the whole surface.

I am not at all convinced that this is well captured by the approach outlined in the paper.

2) Pg. 3, first column: "For a given carpet architecture [...]". The sentence is missing a word, I think.

3) In Fig.2, it is not clear from the paper that the the z-dependent MSDs have been measured in the limit of small velocity. One needs to go search it up in the methods (Sec.3c), where the key assumption is made ($\langle \Delta r^2 \rangle \ll r_0^2$). It would be helpful if this information was more explicit within the main text.

This approximation can be understood in terms of a limit on the observation time for the MSD. Within this limit, however, the MSD should go from ballistic to diffusive, otherwise the approach breaks down (the main local behaviour of the tracer particle becomes advective). It is unclear to me whether this approximation is valid for parameter values typical of experiments with microswimmers, and -if so- down to which distance from the wall is this an acceptable approximation? It would be good to discuss this point in the paper.

4) It is not clear in the paper what results pertain to the rotlet case?

5) In various figures, the axes labels have a rather unusual position. This makes the figures difficult to read. I understand that this strategy might have been devised to save space, but it has made the figures more difficult to read. Please revert to a normal axis labelling.

6) The manuscript should state more explicitly that the thermal diffusivity of the tracers has not been considered in the paper. This is a detail that is important but can be easily overseen by the reader. Further on this, it would be good to add a discussion on the sizes of tracers for which these phenomena can be expected to be relevant. The comparisons with typical experimental parameters are done always with rather large particles in mind (\sim micron sized), but there are plenty of cases where the relevant objects are smaller (e.g. viruses).

Marco Polin

Reviewer #2 (Remarks to the Author):

I have written a report on this paper for another journal. Since the present version of the manuscript is essentially identical to the one I reviewed before, I simply copy my earlier report.

The authors study the dynamics of particles subject to random flow fields generated by different types of actuators (located at an interface). Specifically, they calculate the following:

- Using a 'Stokeslet' approximation they calculate the flow fields generated by different type of actuators at large distances from an interface ('active carpet').
- They calculate the leading moments of these flow fields for a spatial arrangement of actuators that have a uniform distribution in both position and orientation.
- Using these flow fields, which additionally show temporal fluctuations due to (correlated) noise in the force amplitude of the actuators, they determine the statistics of the trajectories of (passive) tracer particles. Averages are performed over different realizations of the temporal noise and different realizations of the distribution of actuators.
- Their analysis results in effective diffusion coefficients of tracer particles that depend on space, i.e. distance to the interface.
- They use Schnitzer's one-dimensional model to analyze the effect of spatially dependent particle speeds and 'collision' rates on macroscopic particle currents. Here they use their results for spatially dependent diffusion constants and assume that the 'collision' rates also show a spatial dependence. The resulting particle currents deviate from Fick's law.

The reasoning of the authors' is flawed at a critical point of their analysis. The actual problem they intend to study is the dynamics of a tracer particle in a flow field generated by different type of actuators, given a random but *fixed* distribution of their positions and orientations. This is a 'quenched disorder' problem, where the variations are sample-to-sample variations. Within a given sample (specific biological realization of an 'active carpet') there are no fluctuations due to the random distribution of the actuators' position and orientation. The variances calculated by the authors (listed in Table I) apply to the typical differences between different samples. It is not clear how averaging over these sample-to-sample variations should inform about the dynamics for a given system. For example, in calculating the position dependence of the diffusion constants they average over both the temporal noise (Ornstein-Uhlenbeck process) and the spatial arrangement of their actuators. What they actually should have done is to study the dynamics by averaging only with respect to the temporal noise and in this case then analyze the resulting stochastic dynamics of the tracer particles. For the applications they have in mind, either in biology or in biotechnology, this would be the relevant analysis. I am afraid that by additionally averaging over the random position and orientation of the actuators they 'add' spurious fluctuations to the particles' dynamics that are absent in an actual system. In their analytical calculations the variance in position of the tracer particles is proportional to the variance in the velocity due to sample-to-sample variations. I also recommend that the authors consult the relevant literature on the dynamics of particles in random velocity fields.

The analysis using Schnitzer's model is not new. The main insight that Fick's law is modified if there are spatial dependencies in a particle's velocity and tumbling/collision rate can already be found in Schnitzer's work and other subsequent work. (From that perspective I find the title and abstract misleading). Furthermore, the authors perform their analysis using Schnitzer's model by assuming space-dependent velocities as well as space-dependent 'collision' rates. The latter is ad hoc and not justified by the analysis in their work. Hence, there is some disconnect between this analysis and the earlier analysis in the authors' manuscript. Moreover, and more importantly, in order to show deviations from Fick's law for 'active carpets' the actual analysis required would be to simulate the dynamics of Brownian particles in a random flow field whose spatial disorder is 'quenched' and whose temporal disorder is 'annealed'. This is not what the authors do, and therefore I would consider their claim that there are deviations from Fick's law for particles in flow fields generated by active carpets to be unfounded.

Given these shortcomings, I cannot recommend the paper for publication.

Reviewer #3 (Remarks to the Author):

The present manuscript analyses in depth the transport of particles generated by active carpets, represented as random distributions of force or other singularities along a planar boundary. This model is relevant to many biological settings where flow is actuated by the motion of various organelles or microorganisms such as ciliary carpets or surface-trapped bacteria. For each type of forcing (i.e. singularity), numerical simulations (and analytical derivations) are performed for a large number of random distribution of singularities along the surface to quantify hydrodynamic fluctuations in terms of the flow velocity variances (there is no net flow for homogeneous distribution of actuators). For time-fluctuating forcings, this can be recast as a spatially-dependent diffusion due to the non-uniform hydrodynamic fluctuations in the direction orthogonal to the active surface. The authors then formulate modified Fick's laws that allow to solve for the concentration of tracer particles in various applications such as sedimentation or diffusion toward a sink in the presence of the hydrodynamic forcing of the active carpets.

I much enjoyed reading and reviewing this manuscript, and am happy to recommend its publication. I only have a few minor comments that the authors may want to consider during the revision of their work.

1) In order to fit in the formatting guidelines of the journal, the reader is asked to continuously go back and forth between the result and methods section. In a more specialised journal, an expanded format of the main text would allow a more linear presentation of the methodology and results. I do not believe that this is a fundamental issue — and I believe that the results presented here are of interest to a wide enough community to warrant publication in Nat. Comm. Yet the authors may want to help the reader in that regard, potentially by including more details in the results section to leave out to the Methods section only the most technical parts.

2) The generalised Fick's laws are formulated in terms of two quantities, namely the variance of the vertical flow v and the memory time of the distribution of actuators, τ . While the algebraic spatial dependence considered in the manuscript is well motivated by the calculations presented in the first part of the manuscript, the motivation of such algebraic dependence for the memory time seems a bit more obscure. Could the authors elaborate a bit more on its origin (e.g. application to sedimentation)?

3) page 6: in analysing the problem of diffusion from a source to an active carpet sink, the authors note that the flux scales quadratically with the force magnitude. Is there a qualitative (or quantitative) argument that could provide some insight on the origin of this result?

4) The generalised Fick's laws are obtained in terms of a single variable z as the problem is invariant in the horizontal direction. In the conclusion, the authors mention a potential generalisation to more complex geometries where such invariance is not retained. Could the authors provide some more elements regarding the generalisation of the modified Fick's laws to three dimensions?

We would like to thank the reviewers for carefully reviewing our manuscript. Here we address the raised comments point by point. We also provide a new version of the manuscript where all changes are marked in blue.

REVIEWER 1:

The paper studies the transport properties of so-called active carpets, collections of agents above a no-slip surface, that stir the surrounding fluid. Active carpets generate stochastic flows in the fluid, and this can affect how suspended particulates are transported. The authors study a simplified version of active carpets, where active agents at low Reynolds number are represented in their far field limit by suitable multipoles derived from the Blake tensor at a distance h from the boundary. The active agents can be either moving randomly along the surface, with a given rotational diffusivity, or are fixed and exert a random perpendicular force which follows an Ornstein-Uhlenbeck process of zero mean. The authors evaluate the statistical properties of the local velocity PDF using an approximation of the flow fields for distances large compared to h , and use these to derive a Fokker-Planck equation for the ensuing transport of passive tracers. They then analyse two test cases, particle sedimentation and diffusion to an active sink, which show clear differences from the thermal equilibrium case. The former shows a clearance region just above the active carpet layer, while the latter can show an increase in the absorbed flux if the source is sufficiently close to the active carpet.

Overall the paper is well written, and the results presented are of high quality (but see a few points raised below). The approach is simple and elegant, and easy to follow. Active matter -both alive and synthetic- has been at the centre of a large research effort over the last years. The current paper is an excellent contribution to the general question of how activity impacts on transport at the micro-scale. Therefore, I am happy to recommend the paper for publication, once the authors address satisfactorily the points raised below.

We are pleased that the referee finds our results of high quality and is happy to recommend publication once the points raised below are addressed. We hope that our new results and further analysis will answer these questions, and that they have improved the paper overall.

1) The modelling approach is based on local average properties of the stochastic flows generated by the active agents. The average here is intended to be with respect to configurations of the active carpet. For agents that meander along the surface, like bacteria, this is justified as long as the timescales involved in the processes (e.g. nutrient absorption) are long compared to the autocorrelation time of the swimmers configuration. This, however, cannot be the case for fixed spatial configuration of the active carpet. In this case, the spatial heterogeneity in the flow field can be frozen in place, and a spatial average would not be equal to a local temporal average. As a result, some parts of the carpet should get a much higher flux of tracers than others. I believe that this can cause a very significant difference between a hypothetical experiment and the theory proposed in this paper.

Let me try to be more explicit. Let us consider for example the case of a carpet of Vorticella, which seems to me to be the experimental system that the authors have in mind when dealing with perpendicular Stokeslets. In this case: i) the cells are fixed in position; ii) the point force that they exert can change in magnitude but is always towards the wall. In this case, although the space averaged flow is still zero, the individual cells should get a massive increase in nutrient uptake due to direct advection towards their mouth. In the end, it is this flux of nutrients that is important for the organism, not the global flux towards the whole surface. I am not at all convinced that this is well captured by the approach outlined in the paper.

To address this important question explicitly, let us consider the referee's example of perpendicular Stokeslets that fluctuate about a non-zero mean. Then, the Ornstein-Uhlenbeck process becomes $\frac{df}{dt} = -\frac{f-\bar{f}}{\tau} + \sigma\eta(t)$, where the mean force is $\langle f \rangle = \bar{f}$ and its variance is $\text{Var}(f) = \sigma^2\tau/2$ as before. The resulting flow is then described by an advective contribution, \mathbf{v}_{adv} due to the mean force, \bar{f} , and a diffusive contribution, \mathbf{v}_{diff} due to its variance, $\text{Var}(f)$. The mean

FIG. S1. *Comparison of advective and diffusive transport* by an active carpets composed of a square lattice of perpendicular Stokeslets. The actuators are located at $(i\lambda, j\lambda, h = 1)$ where i, j are integer numbers. They are all oriented down towards the surface with the same (negative) force F_0 . **A, B.** Flows [Eq. R1] produced by a sparse carpet of $\lambda = 10$ and a dense carpet of $\lambda = 2.5$, respectively, in the plane $y = 0$. Colours show the flow magnitude and black arrows are streamlines. **C.** The total advection flow, $v_{\text{adv},z}(x, 0, 2.5)$, normalised by the flow of a single actuator, $u_{\perp,z}(0, 0, 2.5)$, for different lattice spacings: $\lambda = 10$ (blue), $\lambda = 2.5$ (green), $\lambda = 1$ (red). The flows vanish as λ decreases. **D.** Comparison of advective and diffusive transport. Black points show the normalised advection flow, $\Phi(\zeta)$ [Eq. R2]. The dashed lines are the decaying functions $\exp(-\zeta)$ (dashed green), $\exp(-\zeta^2)$ (dashed blue), so the advection vanishes for dense carpets and large distances from the surface. However, the normalised diffusive transport [Eq. R3] increases with ζ (red lines). **E.** Mean squared displacement, simulated exactly as in Fig. 2A-D, but for a carpet composed of a square lattice of perpendicular Stokeslets with very weak fluctuations, $\text{Var}(f) = 10^{-6}$, compared to a strong mean force directed towards the carpet, $\bar{f} = -1$. The other parameters used are density $n = 1$, $h = 1$, $\tau = 0.1$. The red dashed lines show the prediction of Eq. 3. **F.** Corresponding space-dependent diffusivity. The solid lines show the prediction of Eq. 4. Despite the strong advection currents near the carpet, the theory still holds beyond a certain distance from the surface.

of the diffusive contribution vanishes when averaging over the temporal noise. In the previous version of the paper we wrote that the mean of the advective contribution also vanishes, that $\langle \mathbf{v} \rangle = 0$ when $F(\mathbf{r}_a, \mathbf{p}_a)$ is constant, which is only true for a perfectly uniform carpet. Indeed, the referee is correct that any natural carpet is likely to feature some heterogeneity in its force distribution that can drive local advection flows. The key question is then how strong these local flows are compared to the enhanced diffusion.

Naturally, the advection is not significant in the limit of a small mean force, when $\bar{f}^2 \ll \text{Var}(f)$. Even when the mean force is comparatively large, however, the diffusion can still dominate far from the surface, depending on the structure of $F(\mathbf{r}_a, \mathbf{p}_a)$. This is explained in terms of the local heterogeneities becoming less important when $z \gg r_{\text{nm}}$,

where the typical nearest-neighbour distance between actuators is $r_{\text{nn}} \sim 1/\sqrt{n}$. To quantify this carefully, we consider a square lattice of perpendicular Stokeslets with lattice spacing $r_{\text{nn}} = \lambda$. That is, the forces are located at position $(i\lambda, j\lambda, h)$ where i and j are integer numbers, so the number density $n = 1/\lambda^2$. The total advection generated by this active carpet is given by

$$\mathbf{v}_{\text{adv}}(x, y, z) = \sum_{i=-\infty}^{\infty} \sum_{j=-\infty}^{\infty} \mathbf{u}_{\perp}(x - i\lambda, y - j\lambda, z, h, \bar{f}), \quad (\text{R1})$$

where \mathbf{u}_{\perp} is the flow due to an individual Stokeslet oriented perpendicular to the no-slip surface. This total flow is shown in Fig. S1 for different lattice spacings, where all Stokeslets have the same (negative) force \bar{f} . In all cases, there is a down-welling region (downward flow) near the Stokeslets and, by incompressibility, up-welling regions between the Stokeslets. Perhaps counter-intuitively, at a given distance z from the surface, the sparse carpets (Fig. S1A, with large λ) drive stronger flows than the dense carpets (Fig. S1B, with small λ). This is highlighted in Fig. S1C, which plots the vertical flow velocity along the line $y = 0$ for different values of λ . These curves show the down-welling regions around $x = 0, \pm\lambda, \dots$ and up-welling regions around $x = \pm\lambda/2, \pm3\lambda/2, \dots$, but their amplitude decreases strongly with decreasing λ , i.e. with increasing number density n . This is quantified further in Fig. S1D, which shows the vertical flow directly above a Stokeslet ($x = y = 0$). Using equation M9, we write the normalised total vertical flow as

$$\Phi(\zeta) = \frac{v_{\text{adv},z}}{u_{\perp,z}} \Big|_{x=y=0} = \sum_{i=-\infty}^{\infty} \sum_{j=-\infty}^{\infty} \frac{2 - 3(i^2 + j^2)\zeta^{-2}}{2(1 + (i^2 + j^2)\zeta^{-2})^{7/2}}, \quad (\text{R2})$$

where the dimensionless number $\zeta = z/\lambda = z\sqrt{n}$ and the normalisation factor is $u_{\perp}^z(0, 0, z) = 12h^2\bar{f}/z^3$. Recall that \bar{f} has units m^2/s because forces are scaled with the fluid viscosity. Then, in the limit $\zeta \rightarrow 0$ we recover the flow due to a single Stokeslet, $\Phi \rightarrow 1$, as expected. However, in the limit $\zeta \rightarrow \infty$ the flow tends to zero because the spatial gradients in the actuator density disappear. This decay is quite strong (Fig. S1D; black points), approximately like a Gaussian function, $\Phi(\zeta) \approx \exp(-\zeta^2)$ (dashed blue line). **Thus, the normalised advective transport decays rapidly with ζ , while the diffusive transport actually increases.** Specifically, using $\langle v_{\text{diff},z}^2 \rangle = 15\pi n h^4 \text{Var}(f)/z^4$, we write the normalised diffusive transport as

$$\frac{\sqrt{\langle v_{\text{diff},z}^2 \rangle}}{|u_{\perp,z}|} = \zeta \sqrt{\frac{15\pi \text{Var}(f)}{48 f^2}}, \quad (\text{R3})$$

which is shown in Fig. S1D as red lines. The relative importance of the diffusive and the advective transport is

$$\frac{\sqrt{\langle v_{\text{diff},z}^2 \rangle}}{|v_{\text{adv},z}|} = \frac{\zeta}{\Phi(\zeta)} \sqrt{\frac{15\pi \text{Var}(f)}{48 f^2}}. \quad (\text{R4})$$

Hence, the diffusion dominates over advection beyond a distance $z^* = \zeta^*/\sqrt{n}$ from the carpet, where

$$\zeta^* = \sqrt{\frac{1}{2} W_0 \left(\frac{32 f^2}{5\pi \text{Var}(f)} \right)}, \quad (\text{R5})$$

in terms of the Lambert W_0 function. This occurs at $\zeta^* \approx 0.85$ for $\text{Var}(f) = \bar{f}^2$, which is fairly close to the active carpet. Even when their fluctuations are a thousand times weaker [Fig. S1D; red dotted line], for $\bar{f}^2 = 10^6 \text{Var}(f)$, the transition occurs at $\zeta^* \approx 2.55$, which is still not that far from the carpet. This is especially relevant for high actuator densities. Indeed, many organisms like *Vorticella* colonies can grow fairly dense, approaching close packing.

Another point to note is that **the advective flows can average out in time**: Consider a particle located in a down-welling region, slowly moving down towards an actuator. If the particle is also subject to fluctuations, it can diffuse horizontally into an up-welling region, so it can escape. To demonstrate this, we repeat our diffusion simulations (cf. Fig. 2A-D) for a carpet of perpendicular Stokeslets with very weak fluctuations, $\text{Var}(f) = 10^{-6}$, compared to a strong mean force directed towards the carpet, $\bar{f} = -1$. As shown in Fig. S1E, the mean-squared displacement (MSD) still transitions to diffusive motion when $t > \tau$, and the ballistic advective motion disappears over time. Moreover, the resulting space-dependent diffusivity (Fig. S1F) still agrees with the theoretical prediction (Eq. 4) beyond a distance $z \gtrsim 3$ for all components of D_{ij} .

The nutrient flux that individual organisms receive therefore depends strongly on the distance of the nutrient source. If the source is located at $H < z^*$, then the flux to individual organisms can be large due to advection, as the referee indicated. If the source is located at $H > z^*$, then the flux is determined by diffusion. This spreads out the nutrients horizontally before they reach the surface, so on average all organisms receive the same global diffusive flux.

The theory can also be extended for situations where the relation $\langle v_{\text{diff}}^2 \rangle \ll v_{\text{adv}}^2$ does not hold. This may be important for scenarios in biology or synthetic carpets of intermediate actuator densities. First, as described in the new version of the paper, the generalised Fick's law can be extended to three dimensions:

$$\mathbf{J}(\mathbf{r}) = \mathbf{v}_d \varphi - (\tau \mathcal{V} \cdot \nabla) \varphi - \frac{\varphi \tau}{2} \nabla \cdot \mathcal{V}, \quad (\text{R6})$$

where the variance tensor $\mathcal{V}_{ij} = \langle v_i v_j \rangle$ for $i \in (x, y, z)$, as provided in Table I. Then, one could explicitly insert the advection flow as $\mathbf{v}_d = \mathbf{v}_{\text{adv}}(\mathbf{r})$ into this 3D generalised flux, assuming that the fluctuations are still uniformly distributed on the surface. This advection term could be written in terms of Stokeslets, or found with any other hydrodynamic technique such as the boundary-element method, a squirmer-like model, or computational fluid dynamics (CFD) simulations. The disadvantage of this formulation is that it is inherently system-specific. The advantage is that any flow pattern of interest can be inserted (e.g. ciliary transport, filter feeding, bacteria on surfaces), so the resulting advection-diffusion equations can be solved accordingly.

To summarise, we expect our theory to be consistent with experimental realisations of active carpets, as long as the above considerations of locally heterogeneous carpet architectures are taken into account. We have now included a new section in the paper that describes this in detail. We hope that the referee finds this helpful.

2) Pg. 3, first column: For a given carpet architecture []. The sentence is missing a word, I think.

Thank you pointing out this typo. We have corrected it now.

3) In Fig. 2, it is not clear from the paper that the the z-dependent MSDs have been measured in the limit of small velocity. One needs to go search it up in the methods (Sec.3c), where the key assumption is made, $\langle \delta r^2 \rangle \ll r_0^2$. It would be helpful if this information was more explicit within the main text. This approximation can be understood in terms of a limit on the observation time for the MSD. Within this limit, however, the MSD should go from ballistic to diffusive, otherwise the approach breaks down (the main local behaviour of the tracer particle becomes advective). It is unclear to me whether this approximation is valid for parameter values typical of experiments with microswimmers, and -if so- down to which distance from the wall is this an acceptable approximation? It would be good to discuss this point in the paper.

The referee is absolutely correct that the MSDs in Fig. 2 have been simulated with small flow velocities, such that the condition

$$\langle \delta r^2 \rangle \ll r_0^2 \quad (\text{R7})$$

holds true. This is just a method to make sure that we measure the local diffusivity, $D(z)$ with small variations in z . Once this local diffusivity is determined from the MSDs within this limit, the global stochastic dynamics of particles leaving this local area (with large variations in z) can be solved using the space-dependent Fick's laws, as shown in figures 3 and 4. However, as the referee indicated, this condition imposes a limit on the validity of the analytical model close to the surface. Rewriting Eq. R7 as $z^2 \gg 2D(z)t = 2\tilde{D}t/z^{\alpha+\beta}$, we can rearrange this for a temporal condition, $t/\tau \ll z^{2+\alpha+\beta}/(2\tilde{D}\tau)$. For the particle motion to transition from ballistic to diffusive motion, we also require that $t \gg \tau$ (see Eq. M27b). In other words, the theory is only expected to hold when the timescale of diffusive transport is slow compared to the timescale of the hydrodynamic fluctuations themselves. This is true far from the surface, where

$$z \gg (2\tilde{D}\tau)^{1/(2+\alpha+\beta)}. \quad (\text{R8})$$

For the case of perpendicular Stokeslets, for example, this becomes $z \gg \sqrt[6]{30\pi n h^4 \tau^2 \text{Var}(f)}$. Inserting the values used for simulations in figure 2A-D, being $n = 0.1$, $h = 1$, $\text{Var}(f) = 9/16$, $\tau = 0.1$, we find the condition $z \gg 0.61$. Indeed, beyond this distance we find a good agreement between the simulations and the theory. We have now included this important point in throughout the paper.

4) It is not clear in the paper what results pertain to the rotlet case?

The Stokes rotlet represents torque-generating actuators. In nature, a good example could be a carpet of tethered bacterial flagella or nodal cilia [1] that have a beat form in the shape of a cone pointing down to the surface. As the actuators move around in circles in the xy plane, a torque is applied to the liquid. Another example could be a carpet of more common cilia, beating almost perfectly in a plane [2], but with some off-plane fluctuations. For such situations one could combine different terms from the multipole expansion (parallel Stokeslet, rotlets, and perhaps high-order terms) to give a better realistic description. Another example outside biology one could think of is a synthetic active carpet with fluctuating rotors.

The Stokes rotlet only generate flows in the xy plane, but not in the vertical direction (see new Methods section 1). Consequently, also the variance and diffusion due to rotlets only have horizontal components. This fact could be exploited to tune the diffusion anisotropy, the relative magnitude of D_{xx} and D_{zz} . For example, one could use an active carpet made of both Stokeslets and rotlets, and vary their relative prefactors f_{\perp} and ρ , or relative densities n_{\perp} and n_{ρ} . We have clarified this now. In Methods §1, we have also written out this flow field explicitly.

5) In various figures, the axes labels have a rather unusual position. This makes the figures difficult to read. I understand that this strategy might have been devised to save space, but it has made the figures more difficult to read. Please revert to a normal axis labelling.

Of course, we have corrected this now.

6) The manuscript should state more explicitly that the thermal diffusivity of the tracers has not been considered in the paper. This is a detail that is important but can be easily overseen by the reader. Further on this, it would be good to add a discussion on the sizes of tracers for which these phenomena can be expected to be relevant. The comparisons with typical experimental parameters are done always with rather large particles in mind (micron sized), but there are plenty of cases where the relevant objects are smaller (e.g. viruses).

Overall we find that the enhanced diffusion can be much larger than thermal diffusion, but of course this effect is expected to be less significant for very small objects. To quantify this, we consider the fraction

$$\frac{D_{zz,f_{\perp}}}{D_{th}} = \frac{15\pi n h^4 \text{Var}(f) \tau}{z^4} / \frac{k_B T}{6\pi \mu a_p}, \quad (\text{R9})$$

where we used the Stokes-Einstein relation for the thermal diffusivity D_{th} . Equating this expression to unity and solving for the particle size, we find that the active diffusion dominates for particles much larger than

$$a_p^* = \frac{k_B T z^4}{90\pi^2 n h^4 \mu \text{Var}(f) \tau}. \quad (\text{R10})$$

Inserting typical values for *Vorticella* with $h \sim 150\mu\text{m}$, $n \sim 1/(100\mu\text{m})^2$, $\tau \sim 1\text{s}$ and $8\pi\mu f_{\perp} \sim 500\text{pN}$ [3], under standard conditions of viscosity $\mu = 10^{-3}\text{Pa s}$ and temperature $T = 298\text{K}$, we find that even at large distances of $z = 1\text{mm}$ we have $a_p^* \sim 1$ Angstrom. **So, we expect our results to be relevant even for the smallest particles, including viruses and nutrient molecules.**

Indeed, we can also solve Eq. R9 for the distance below which the active diffusion dominates. That gives

$$z_{th} = \sqrt[4]{15\pi n h^4 \langle f^2 \rangle \tau / D_{th}}. \quad (\text{R11})$$

For a small molecule of thermal diffusivity $D_{th} \sim 10^{-9}\text{m}^2/\text{s}$, we find $z_{th} \sim 1\text{mm}$ from the surface. That is much larger than the cell size. Moreover, for micron-sized prey of $D_{th} \sim 0.5\mu\text{m}^2/\text{s}$, we find $z_{th} \sim 7\text{mm}$, which is orders of magnitude larger than the organism itself. We have now discussed this in the paper.

Additionally, the thermal diffusion can also be included explicitly in the generalised Fick's laws. For example, when considering sedimentation near an active carpet, we have

$$J_z(z) = -v_g \varphi - D_{th} \frac{\partial \varphi}{\partial z} - \frac{\tilde{D}}{z^{\alpha+\beta}} \frac{\partial \varphi}{\partial z} + \frac{\alpha \tilde{D}}{2z^{\alpha+\beta+1}} \varphi. \quad (\text{R12})$$

Then, the expression $J_z = 0$ can still be solved analytically to determine the steady-state sedimentation profile. For parallel Stokeslets, for example, with $\alpha = 2$ and $\beta = 0$, we find the solution

$$\frac{\varphi(z)}{\varphi_0} = \frac{z}{\sqrt{\tilde{D} + D_{th} z^2}} \exp \left(\sqrt{\frac{\tilde{D}}{D_{th}^3}} v_g \tan^{-1} \left(\frac{\sqrt{D_{th} z}}{\sqrt{\tilde{D}}} \right) - \frac{v_g z}{D_{th}} \right). \quad (\text{R13})$$

It is important to note that this function has the same shape as the original solution without thermal diffusion. Indeed, particles are still repelled from the active surface, $\lim_{z \rightarrow 0} \varphi(z) = 0$, and the function has a maximum at the same location as before, at $z_{\max} = (\tilde{D}/v_g)^{1/3}$ for all $D_{th} \geq 0$. Thus, **the self-cleaning effect is not affected by thermal diffusion**. Conversely, when the surface activity vanishes, $\tilde{D} \rightarrow 0$, we recover the Boltzmann distribution. We have included this new result in the paper and a new section Methods §5c.

Similarly, the diffusive flux from a source to a sink can also be computed in the presence of both thermal and active fluctuations. As before, we must solve $\partial_z J_z = 0$ for the flux (Eq. R12) without gravity, with boundary conditions $\varphi(0) = 0$ and $\varphi(H) = \varphi_+$. For parallel Stokeslets this yields the concentration profile

$$\frac{\varphi(z)}{\varphi_+} = \frac{z \left(\sqrt{\tilde{D} + D_{th}H^2} - \sqrt{\frac{\tilde{D}(\tilde{D} + D_{th}H^2)}{\tilde{D} + D_{th}z^2}} \right)}{H \left(\sqrt{\tilde{D} + D_{th}H^2} - \sqrt{\tilde{D}} \right)}, \quad (\text{R14})$$

and the corresponding diffusive flux

$$\frac{J_z}{\varphi_+} = - \left(\sqrt{\tilde{D}^2 + D_{th}\tilde{D}H^2} + \tilde{D} + D_{th}H^2 \right) H^{-3}. \quad (\text{R15})$$

As expected, in the limit $\tilde{D} \rightarrow 0$ we recover the thermal flux, $J_z = -D_{th}\varphi_+/H$. This corresponds to ~ 50 particles/second for molecular diffusion with $D_{th} \sim 500\mu\text{m}^2/\text{s}$, and using $\varphi_+ \sim 1$ particle/ μm and $H \sim 10\mu\text{m}$. Conversely, in the limit $D_{th} \rightarrow 0$ we recover the original solution, $J_z = -2\tilde{D}\varphi_+/H^3$. This gives a ‘bare’ active flux of ~ 60 particles/second when inserting the typical values $h \sim 1\mu\text{m}$, $n \sim 1\mu\text{m}^{-2}$, $D_r \sim 1\text{s}^{-1}$ and $8\pi\mu f_{\parallel} \sim 1\text{pN}$. Interestingly, these thermal and active fluxes do not just add up, because there is also a cross term. Indeed, the total flux from Eq. R15 gives $J_z \sim 128$ particles/second. Therefore, **the thermal diffusion can actually enhance the active diffusive flux, and vice versa, since they co-operate with one another**. We have included this discussion in the paper.

REVIEWER 2:

The authors study the dynamics of particles subject to random flow fields generated by different types of actuators (located at an interface). Specifically, they calculate the following: -Using a ‘Stokeslet’ approximation they calculate the flow fields generated by different type of actuators at large distances from an interface (‘active carpet’). -They calculate the leading moments of these flow fields for a spatial arrangement of actuators that have a uniform distribution in both position and orientation. -Using these flow fields, which additionally show temporal fluctuations due to (correlated) noise in the force amplitude of the actuators, they determine the statistics of the trajectories of (passive) tracer particles. Averages are performed over different realizations of the temporal noise and different realizations of the distribution of actuators. -Their analysis results in effective diffusion coefficients of tracer particles that depend on space, i.e. distance to the interface. -They use Schnitzer’s one-dimensional model to analyze the effect of spatially dependent particle speeds and ‘collision’ rates on macroscopic particle currents. Here they use their results for spatially dependent diffusion constants and assume that the ‘collision’ rates also show a spatial dependence. The resulting particle currents deviate from Fick’s law.

The reasoning of the authors’ is flawed at a critical point of their analysis. The actual problem they intend to study is the dynamics of a tracer particle in a flow field generated by different type of actuators, given a random but *fixed* distribution of their positions and orientations. This is a ‘quenched disorder’ problem, where the variations are sample-to-sample variations. Within a given sample (specific biological realization of an ‘active carpet’) there are no fluctuations due to the random distribution of the actuators’ position and orientation. The variances calculated by the authors (listed in Table I) apply to the typical differences between different samples. It is not clear how averaging over these sample-to-sample variations should inform about the dynamics for a given system. For example, in calculating the position dependence of the diffusion constants they average over both the temporal noise (Ornstein-Uhlenbeck process) and the spatial arrangement of their actuators. What they actually should have done is to study the dynamics by averaging only with respect to the temporal noise and in this case then analyze the resulting stochastic dynamics of the tracer particles. For the applications they have in mind, either in biology or in biotechnology, this would be the relevant analysis. I am afraid that by additionally averaging over the random position and orientation of the actuators they ‘add’ spurious fluctuations to the particles’ dynamics that are absent in an actual system. In their analytical calculations the variance in position of the tracer particles is proportional to the variance in the velocity due to sample-to-sample variations. I also recommend that the authors consult the relevant literature on the dynamics of particles in random velocity fields.

We are very surprised by this strong criticism. There seems to be a critical misunderstanding about the methodology. The referee writes: *“What they actually should have done ...relevant analysis”*. We are surprised by this statement, because we do perform the relevant analysis. Perhaps the referee has not noticed that in all our simulations, we fully resolve the positions and orientations of all the discrete active agents throughout time. Then, we determine all the key quantities (enhanced diffusivities, sedimentation profiles, nutrient fluxes), without any prior averaging over the positions or orientations of the active agents. Therefore, any quenched disorder (e.g. from fixed carpet configurations) is explicitly included in these simulations, and **all our main findings are unequivocally supported by this data throughout the paper**: (1) That active carpets made of biological and engineered actuators can drive an enhanced diffusion much larger than thermal fluctuations; (2) That a “self-cleaning” effect can occur for particles sedimenting towards an active carpet; and (3) That the diffusive (nutrient) flux from a source to an active sink can be significantly larger than the thermal flux. As highlighted by the other referees, these results are of high quality and relevant to many biological settings.

The only question is then whether the theoretical model can also predict these results accurately. The short answer is that the theory already agrees well with all the simulations presented in the paper, but there are also situations where the theory does not provide a satisfactory approximation. That is, the model can only predict our results if certain conditions hold true. We apologise that these conditions were not clearly stated earlier.

In particular, the referee is concerned about potential effects of quenched disorder, the notion that a system has random variables that do cannot evolve in time. In our system, the relevant random variables that determine the particle transport are the relative positions between the actuators and the tracer particles. **Even if the actuators themselves are fixed, the tracers may still diffuse in space, so under the right conditions the relative positions are not quenched.** Then, the key question is what these conditions are. The first requirement is that the advective transport (due to individual actuators locally) is much weaker than the diffusive transport (due all

actuators together, aided by thermal noise). For example, if a tracer is caught in a local actuator current, then its dynamics are effectively quenched; but if it can diffuse away and escape, then its dynamics are effectively annealed. As demonstrated in the new Figure S1, this requirement is satisfied if the particles are located a certain distance away from the active surface (see Eq. R4). The second requirement is that the timescale of diffusive transport is slow compared to the timescale of the hydrodynamic fluctuations themselves. This ensures that the tracer motion is diffusive over time and not ballistic according to a specific (possibly fixed) carpet configuration. This requirement is also satisfied if the particles are located a certain distance away from the active surface (see Eq. R8).

In the end, we are interested in computing time-averaged quantities, such as the steady-state sedimentation profiles and nutrient fluxes. As the tracers spread out by diffusion over time, this temporal averaging becomes equivalent to spatial averaging (annealed disorder). Therefore, our theory correctly predicts all the simulations in the paper, also those for fixed carpet configurations. Specifically, the theory is correct if the conditions described above are satisfied, which is the case in many biological and engineered settings. On the one hand, the advantage of this annealed disorder model is that the main results are captured with relatively simple equations. These accessible formulae can readily be implemented by people across the disciplines, from cell biology and physics to material sciences, whilst being careful of the limitations. On the other hand, future theories could perhaps relax these conditions by taking the effects of quenched disorder into account. We expect this could be an exciting opportunity of further research in the field of non-equilibrium statistical mechanics and active matter systems.

On balance, we find that the sentence “*The reasoning of the authors is flawed at a critical point of their analysis*” is a disproportionately strong judgement. The analysis is not flawed, but we apologise that it was not clear when exactly it is applicable. We hope that this response has lightened the referee’s concerns, and that perhaps it has spurred a sense of enthusiasm.

The analysis using Schnitzer’s model is not new. The main insight that Fick’s law is modified if there are spatial dependencies in a particle’s velocity and tumbling/collision rate can already be found in Schnitzer’s work and other subsequent work. (From that perspective I find the title and abstract misleading). Furthermore, the authors perform their analysis using Schnitzer’s model by assuming space-dependent velocities as well as space-dependent ‘collision’ rates. The latter is ad hoc and not justified by the analysis in their work. Hence, there is some disconnect between this analysis and the earlier analysis in the authors’ manuscript. Moreover, and more importantly, in order to show deviations from Fick’s law for ‘active carpets’ the actual analysis required would be to simulate the dynamics of Brownian particles in a random flow field whose spatial disorder is ‘quenched’ and whose temporal disorder is ‘annealed’. This is not what the authors do, and therefore I would consider their claim that there are deviations from Fick’s law for particles in flow fields generated by active carpets to be unfounded.

We respectfully disagree with the referee that our results are not new. The whole concept of surface-driven hydrodynamic fluctuations by active carpets has never been described before, not in any existing paper. **The main results are unique, and they are directly relevant to scientists in many fields;** For example to predict enhanced nutrient fluxes in ecology, or to design self-cleaning active coating materials. Additionally, our paper also leaves a new challenge to statistical mechanics, to find the general solution of its complex dynamics out of equilibrium. Therefore, we believe that this work offers an important new contribution, which would benefit the community across these disciplines. Concerning the connection with the paper by Schnitzer [4], we would like to clarify that this (highly respected) paper does not discuss active carpets, hydrodynamic flows or surface-driven fluctuations. It is still directly related to biological statistical mechanics, because it describes the chemotaxis of *E. coli* bacteria. Yet, it does not describe the diffusion of passive particles driven by living cells, but rather the diffusive motion of the bacteria themselves. The referee is of course correct that we use their mathematical machinery to solve for the space-dependent particle diffusion. However, to achieve this, we first had to derive the hydrodynamic fluctuations $\mathcal{V}_{ij}(\mathbf{r}) = \langle v_i v_j \rangle$ and the space-dependent diffusion tensor $D_{ij}(z)$ for the different types of active carpets. Afterwards, we still had to solve the resulting generalised Fick’s laws to determine the enhanced nutrient fluxes and the non-equilibrium sedimentation profiles, and verify all our results with detailed simulations. Thus, the intellectual concepts and key results but also the methodology of our work is distinctly different.

Finally, regarding the title of the paper, we do somewhat agree that the previous title puts too much emphasis on the generalised Fick’s laws without giving due credit to Schnitzer and others in this field, and perhaps too little emphasis on the other aspects of our paper. Therefore, we have changed the title and adapted the abstract accordingly. We hope this is more balanced now.

We thank the referee again for their detailed comments. We think they have really improved the paper.

REVIEWER 3:

The present manuscript analyses in depth the transport of particles generated by active carpets, represented as random distributions of force or other singularities along a planar boundary. This model is relevant to many biological settings where flow is actuated by the motion of various organelles or microorganisms such as ciliary carpets or surface-trapped bacteria. For each type of forcing (i.e. singularity), numerical simulations (and analytical derivations) are performed for a large number of random distribution of singularities along the surface to quantify hydrodynamic fluctuations in terms of the flow velocity variances (there is no net flow for homogeneous distribution of actuators). For time-fluctuating forcings, this can be recast as a spatially-dependent diffusion due to the non-uniform hydrodynamic fluctuations in the direction orthogonal to the active surface. The authors then formulate modified Ficks laws that allow to solve for the concentration of tracer particles in various applications such as sedimentation or diffusion toward a sink in the presence of the hydrodynamic forcing of the active carpets.

I much enjoyed reading and reviewing this manuscript, and am happy to recommend its publication. I only have a few minor comments that the authors may want to consider during the revision of their work.

We are glad that the referee enjoyed reading our manuscript, and we are thankful for their positive appraisal overall.

1) In order to fit in the formatting guidelines of the journal, the reader is asked to continuously go back and forth between the result and methods section. In a more specialised journal, an expended format of the main text would allow a more linear presentation of the methodology and results. I do not believe that this is a fundamental issue and I believe that the results presented here are of interest to a wide enough community to warrant publication in Nat. Comm. Yet the authors may want to help the reader in that regard, potentially by including more details in the results section to leave out to the Methods section only the most technical parts.

We agree with the referee that it is very helpful to include more of the methods in the main paper. We have conveyed this question to the editors, who responded as follows:

“Your Article (Introduction, Results, Discussion) can be up to 5000 words/10 display items, and the Methods section without a word-limit, so please incorporate as much of the Supplementary Information into the main paper as possible. When moving material to the main paper, the most important figures, tables etc should be selected.”

Following this advice, we have now included as much of the methods to the main text as possible, whilst making sure that the formatting guidelines are followed appropriately. Similarly, we thoroughly revised the method section itself so that it can be read chronologically as well. We hope this has made the paper much clearer.

2) The generalised Ficks laws are formulated in terms of two quantities, namely the variance of the vertical flow v and the memory time of the distribution of actuators, τ . While the algebraic spatial dependence considered in the manuscript is well motivated by the calculations presented in the first part of the manuscript, the motivation of such algebraic dependence for the memory time seems a bit more obscure. Could the authors elaborate a bit more on its origin (e.g. application to sedimentation)?

We thank the referee for this important question. In general, any functional form of the space-dependent memory time $\tau(z)$ can be used, as long as the assumptions in the derivation of the generalised Fick’s laws are satisfied. One is not restricted to the ansatz $\tau(z) = \tilde{\tau}/z^\beta$, but we chose this algebraic form because it is very common in natural systems. To name a few examples: $\beta = 0$ corresponds to a constant memory time. $\beta = -1$ corresponds to the time-scale $\tau \sim z/v_a$ associated with a swimmer moving underneath a tracer particle. $\beta = -2$ corresponds to the time-scale $\tau \sim z^2/D_a$ associated with actuators diffusing underneath a tracer particle. Note that one could have inverted the exponent to keep β positive, but this does not change anything physically. Indeed, we preferred to keep this expression consistent with the one for the space-dependent velocity fluctuations. We have clarified this now in the paper.

3) page 6: in analysing the problem of diffusion from a source to an active carpet sink, the authors note that the flux scales quadratically with the force magnitude. Is there a qualitative (or quantitative) argument that could provide some insight on the origin of this result?

This is indeed an interesting question. By symmetry, we expect the nutrient flux to scale quadratically with the force, because the equations are invariant under the transformation $\mathbf{f} \rightarrow -\mathbf{f}$. We have included this in the text now.

4) The generalised Ficks laws are obtained in terms of a single variable z as the problem is invariant in the horizontal direction. In the conclusion, the authors mention a potential generalisation to more complex geometries where such invariance is not retained. Could the authors provide some more elements regarding the generalisation of the modified Ficks laws to three dimensions?

We thank the referee for this exciting question. The analysis of the generalised Fick's laws has now been extended to three dimensions. As described in Methods §4, this gives the three-dimensional flux

$$\mathbf{J}(\mathbf{r}) = \mathbf{v}_d \varphi - (\tau \mathcal{V} \cdot \nabla) \varphi - \frac{\varphi \tau}{2} \nabla \cdot \mathcal{V}, \quad (\text{R16})$$

where the variance tensor $\mathcal{V}_{ij}(\mathbf{r}) = \langle v_i v_j \rangle$ is given in Table I for different actuator types on a planar no-slip surface.

Our theory may also be extended to more complex geometries by taking the following steps: First, one should find the hydrodynamic Green's function (cf. the Blake tensor in Eq. M1) that satisfies the Stokes equations and the boundary conditions of the geometry in question. Once this first step is taken, all results could also be verified with detailed simulations. Second, the mean flow $\langle \mathbf{v}(\mathbf{r}) \rangle$ should be determined by integrating this Green's function over the carpet along with its force distribution, as in Eq. M12. This may tend to zero for homogeneously distributed carpets, depending on the surface shape and the distribution of actuator positions and orientations, but not necessarily. Third, one should determine the variance tensor $\mathcal{V}_{ij}(\mathbf{r})$ as in Eq. M13, which may in general be dependent on all three spatial coordinates. Fourth, the generalised flux may be extended by revisiting Schnitzer's telegraph model, as described in Methods §4. These equations may then be solved numerically or analytically, but care should be taken that the conditions for the theory to be accurate are correctly translated to the new geometry. We have included this interesting discussion in the text.

To conclude, we would like to thank the referees once more for their time and detailed comments.

-
- [1] S. Nonaka, S. Yoshida, D. Watanabe, S. Ikeuchi, T. Goto, W. F. Marshall, and H. Hamada, "De novo formation of left-right asymmetry by posterior tilt of nodal cilia," *PLoS Biol.* **3**, e268 (2005).
 - [2] M. A. Chilvers and C. O'Callaghan, "Analysis of ciliary beat pattern and beat frequency using digital high speed imaging: comparison with the photomultiplier and photodiode methods," *Thorax* **55**, 314–317 (2000).
 - [3] R. E. Pepper, M. Roper, S. Ryu, N. Matsumoto, M. Nagai, and H. A. Stone, "A new angle on microscopic suspension feeders near boundaries," *Biophys. J.* **105**, 1796–1804 (2013).
 - [4] M. J. Schnitzer, "Theory of continuum random walks and application to chemotaxis," *Phys. Rev. E* **48**, 2553–2568 (1993).

REVIEWERS' COMMENTS

Reviewer #1 (Remarks to the Author):

I have read with interest the authors' comments and I am satisfied with their answers and the amendments made to the paper.

I am now happy to recommend the manuscript for publication.

Reviewer #2 (Remarks to the Author):

We appreciate the authors' efforts to clarify the role of the different types of noise in their system. While the answers they give in their reply to our criticisms are partially satisfactory, the paper still lacks clarity in this respect. In their theoretical calculations they determine variations from sample to sample and call these "hydrodynamic fluctuations". We find this confusing, if not misleading. It is not clear from what is written in the main text how the simulations for fixed actuators performed by the authors are related to their theory. This needs to be greatly improved before publication, and we would recommend that the authors make a considerable effort to do so. Secondly, the authors have now better explained that and how their mathematical analysis agrees with Schnitzer's. As we have noted in our previous report, this part of the analysis is a fairly straightforward application of known results to a particular case that is new and potentially interesting. We are still not enthusiastic about this work, but we would reconsider our opinion if the authors would substantially reformulate their work and clarify the different aspects of randomness that they take into account in their simulations and analytical calculations. In its present form, the manuscript lacks clarity.

Reviewer #3 (Remarks to the Author):

I would like to thank the authors to take the time to respond to my comments in detail and for their implementation in the manuscript. I am satisfied with their response.

We thank the reviewers again for carefully reviewing our manuscript.

REVIEWER 1:

I have read with interest the authors' comments and I am satisfied with their answers and the amendments made to the paper. I am now happy to recommend the manuscript for publication.

We are glad that the referee has read our resubmission with interest, and that the referee now recommends publication.

REVIEWER 2:

We appreciate the authors' efforts to clarify the role of the different types of noise in their system. While the answers they give in their reply to our criticisms are partially satisfactory, the paper still lacks clarity in this respect. In their theoretical calculations they determine variations from sample to sample and call these "hydrodynamic fluctuations". We find this confusing, if not misleading. It is not clear from what is written in the main text how the simulations for fixed actuators performed by the authors are related to their theory. This needs to be greatly improved before publication, and we would recommend that the authors make a considerable effort to do so.

We agree with the referee that the term "hydrodynamic fluctuations" could cause confusion. Therefore, throughout the text, we have renamed this term "active fluctuations", which is defined as the total flows generated by all the actuators that together push and pull particles or molecules located above the active carpet. This is explained in detail on page 3, in the section called 'active fluctuations' (which used to be called 'hydrodynamic fluctuations'). We have also explained this concept carefully throughout the text to avoid any confusion, particularly with other sources of noise like Brownian motion.

Also included are three new supplementary videos that illustrate the concept of active fluctuations.

- Video 1 shows how the fixed actuators (with a force governed by independent Ornstein-Uhlenbeck processes) together lead to a total flow $\mathbf{v}(t)$ that evolves dynamically.
- Video 2 shows the sedimentation of particles towards an active carpet. Far away the force of gravity dominates, but nearby the active fluctuations can repel the particles, leading to a sedimentation profile that does not follow the Boltzmann distribution.
- Video 3 shows diffusion of particles from a source to an active sink. The different colours correspond to the amount of crossings made, and thus relate to the particle flux.

Additionally, on page 4, in the section "space-dependent diffusivity", we have also clarified how the simulations for fixed actuators are related to the theory. Specifically, we explained better how the active fluctuations can vary over time, and here and throughout the text we clarified the role of the different types of noise in our system.

To make the paper more readable, the most important parts of the methods section were moved or described in more detail in the main text. We also moved figure S1 to figure 5, as described on pages 7-8.

Secondly, the authors have now better explained that and how their mathematical analysis agrees with Schnitzer's. As we have noted in our previous report, this part of the analysis is a fairly straightforward application of known results to a particular case that is new and potentially interesting. We are still not enthusiastic about this work, but we would reconsider our opinion if the authors would substantially reformulate their work and clarify the different aspects of randomness that they take into account in their simulations and analytical calculations. In its present form, the manuscript lacks clarity.

We thank the reviewer for noting that part of the analysis is new and potentially interesting. We have now clarified the different aspects of randomness in our simulations and theory. In every section we explicitly state the source of fluctuations that the particles are subject to, be it active or thermal fluctuations or both. We are grateful to the referee for the detailed and valuable comments.

REVIEWER 3:

I would like to thank the authors to take the time to respond to my comments in detail and for their implementation in the manuscript. I am satisfied with their response.

We thank the referee once more for her/his time and efforts.